Automated stratigraphic interpretation from drillhole lithological descriptions with uncertainty
quantification: litho2strat 1.0
Vitaliy Ogarko[1,2] and Mark Jessell[1,2]
[1]Centre for Exploration Targeting (School of Earth Sciences), The University of Western Australia,
Crawley, 6009 WA, Australia
[2]Mineral Exploration Cooperative Research Centre, The University of Western Australia, Crawley,
6009 WA, Australia
Correspondance: Vitaliy Ogarko (vitaliy.ogarko@uwa.edu.au)

# Abstract

Australian commonwealth, state and territory geological surveys possess information on over 3 million drillhole logs. In addition to mineral exploration drilling, extensive drillhole datasets exist from oil and gas exploration and hydrogeological studies. Other countries no doubt have similar data holdings. Together these legacy drillhole datasets have the potential to significantly enhance constraints on regional 3D geological models and improve our understanding of subsurface architecture, but have limited use in their current form as many if not most drill logs lack stratigraphic information, containing only lithological descriptions.

This study develops open-source codes and methodologies for stratigraphy recovery (determining the ordered sequence of stratigraphic units) from drillhole lithological data by introducing a search algorithm that systematically explores all geologically plausible stratigraphic orderings for individual drillholes, combined with a solution correlation algorithm that compares the topological relationships of stratigraphic units across multiple drillholes to identify geologically consistent solutions and reduce uncertainty. The algorithms combine constraints from lithological descriptions with stratigraphic relationships automatically derived from regional maps. In addition, the method quantifies uncertainty by generating multiple plausible stratigraphic interpretations, providing critical insights for resource estimation, scenario analysis, and data acquisition strategies.

The application of our method to a dataset of 52 drillholes from South Australia demonstrated its ability to make useful predictions of stratigraphic solutions and quantifying associated uncertainties. These results not only validate our approach but also highlight opportunities to refine current stratigraphic descriptions and provide a valuable new source for regional 3D geological modelling.

## 1. Introduction

Drillhole data serve as a fundamental constraint for subsurface geological exploration and 3D geological modelling, offering direct insights into lithological and hence stratigraphic features (Wellmann & Caumon, 2018). However, the inherent sparsity of such data, coupled with challenges posed by legacy datasets maintained by industry and Geological Survey Organizations (GSOs), often hinders comprehensive geological understanding and modelling (Jessell et al., 2010; Pakyuz-Charrier et al., 2018). GSOs' databases typically contain lithological information as unstructured text descriptions (e.g., 'sandy limestone with minor shale') but rarely include stratigraphic unit assignments. This creates a critical gap in the data needed for accurate and meaningful geological predictions (Hartmann & Moosdorf, 2012).

Geological modelling plays a crucial role in understanding subsurface structures and processes, providing a foundation for various applications in earth sciences (Jessell et al., 2014). Such modelling commonly relies on datasets such as borehole data, geophysical data, and mapping data. Among these, borehole data provide the most accurate insights into subsurface geology and stratigraphy (Guo et al., 2022). The models generated through geological modelling can serve dual purposes: they can be directly employed for geological interpretations, such as identifying fault systems, and mineral deposits (Alvarado-Neves et al., 2024; Vollgger et al., 2015), or they can be integrated as constraints in methodologies that use a prior 3D model, such as geophysical inversions (Giraud et al., 2017; Martin et al., 2024; Ogarko et al., 2021; Tarantola, 2005) and hydrogeological forward modelling (D'Affonseca et al., 2020).

Modern drillhole measurement techniques primarily focus on chemical, mineralogical and lithological characterization, whereas the fundamental categorical unit of regional 3D geological models is defined by stratigraphy (Calcagno et al., 2008; Caumon et al., 2009; Mallet, 2002). This discrepancy underscores the need for innovative approaches to recover and integrate stratigraphic information from existing datasets.

Recent advancements in automation have made significant progress in processing drillhole data, though most address different aspects of the problem than stratigraphic recovery. Data standardization tools like dh2loop (Joshi et al., 2021) extract and harmonize lithological descriptions from unstructured text using thesauri and fuzzy string matching, providing essential preprocessing for downstream analysis. Pattern recognition methods (Schetselaar & Lemieux, 2012) can identify lithostratigraphic markers and contacts within drill logs, helping to detect boundaries between units. Machine learning approaches for 3D geological modeling (Guo et al., 2024) can interpolate between drillholes to create subsurface models, but typically require pre-interpreted stratigraphic data as input. While these methods provide valuable components of the workflow, none directly address the fundamental challenge of transforming lithological descriptions into stratigraphic interpretations with quantified uncertainties.

Existing automated interpretation methods primarily work with different data types than those available in legacy drillhole databases. Geophysics-based methods (Wu & Nyland, 1987; Fullagar et al., 2004; Silversides et al., 2015) leverage distinctive signatures in gamma, resistivity, or other wireline logs to predict stratigraphic units, but require geophysical data that are absent from most legacy drillholes. Geochemical and spectral approaches (Hill & Uvarova, 2018) use XRF scanning or hyperspectral measurements to identify geological boundaries with high precision, but depend on expensive data acquisition unavailable in historical datasets. Hybrid machine learning methods, such as those applied in the Pilbara iron ore deposits (Wedge et al., 2019), combine lithology with assays

and geophysics but require extensive pre-interpreted drillhole datasets for training, limiting their
application in greenfield exploration areas. These approaches do not address the fundamental
challenge faced by geological surveys worldwide: millions of legacy drillholes contain only lithological
descriptions but lack both stratigraphic interpretations and the geophysical logs required by current
automated methods.
To address these challenges, we formulate the problem of stratigraphic recovery from drillhole
databases as follows. The input to our methodology consists of: (1) legacy drillhole databases
containing lithological descriptions (e.g., "sandstone", "siltstone", "dolomite") at various depth
intervals, typically without stratigraphic labels; (2) regional geological maps that define stratigraphic
unit boundaries and their spatial relationships; and (3) topological constraints that specify which
stratigraphic units can be in contact based on their known relative ages and depositional sequences.
The output comprises: (1) multiple plausible stratigraphic solutions, where each solution provides unit
assignments for all depth intervals in the drillholes; (2) their ranking by geological likelihood; and (3)
quantified uncertainties for these interpretations. The objective is threefold: first, to systematically
transform lithological descriptions into stratigraphic interpretations by testing all geologically plausible
orderings of stratigraphic units that are consistent with the observed lithologies; second, to quantify
the uncertainty inherent in these interpretations given that multiple stratigraphic units may share
similar lithological characteristics; and third, to establish correlations between multiple drillholes to
reduce uncertainty and improve the reliability of stratigraphic assignments across a region. This
transformation is essential because regional 3D geological models are fundamentally organized by
stratigraphy rather than lithology, yet the majority of legacy drillhole data lack stratigraphic labels.
Figure 1 illustrates this challenge with a simplified example: a drillhole log with four lithological
intervals (sandstone, siltstone, sandstone, dolomite) could correspond to multiple stratigraphic
interpretations. The two sandstone intervals might represent the same formation repeated by faulting,
or they could belong to different formations with similar but distinct lithological compositions. Without
additional constraints, both interpretations are geologically plausible, highlighting the inherent
ambiguity in stratigraphic assignment from lithological data alone.

The Challenge of Stratigraphic Interpretation from Lithological Data

**The Challenge:** The same lithology (e.g., "sandstone") can belong to multiple stratigraphic formations with different compositions. Our method systematically explores all geologically plausible assignments, ranks them by likelihood, and quantifies uncertainty.


Figure 1: Schematic illustration of the stratigraphic interpretation problem. A drillhole log containing only lithological descriptions (left) can yield multiple plausible stratigraphic solutions (right) because the same lithology may occur in different stratigraphic formations with varying compositions.

This study develops open-source codes and methodologies for stratigraphy recovery from drillhole lithological data through a two-stage approach. First, we introduce a branch-and-prune search algorithm that systematically explores all geologically plausible stratigraphic orderings for individual drillholes. Second, we apply a solution correlation algorithm that integrates information from multiple drillholes by comparing topological relationships of stratigraphic units, thereby enhancing the robustness and reliability of interpretations. The method quantifies uncertainty by generating multiple plausible stratigraphic interpretations, providing critical insights for resource estimation, scenario analysis, and data acquisition strategies. We apply our method to a dataset of 52 drillholes from South Australia to demonstrate its practical application and validate its performance against existing stratigraphic interpretations.

# 2. Methodology

## 2.1 Workflow

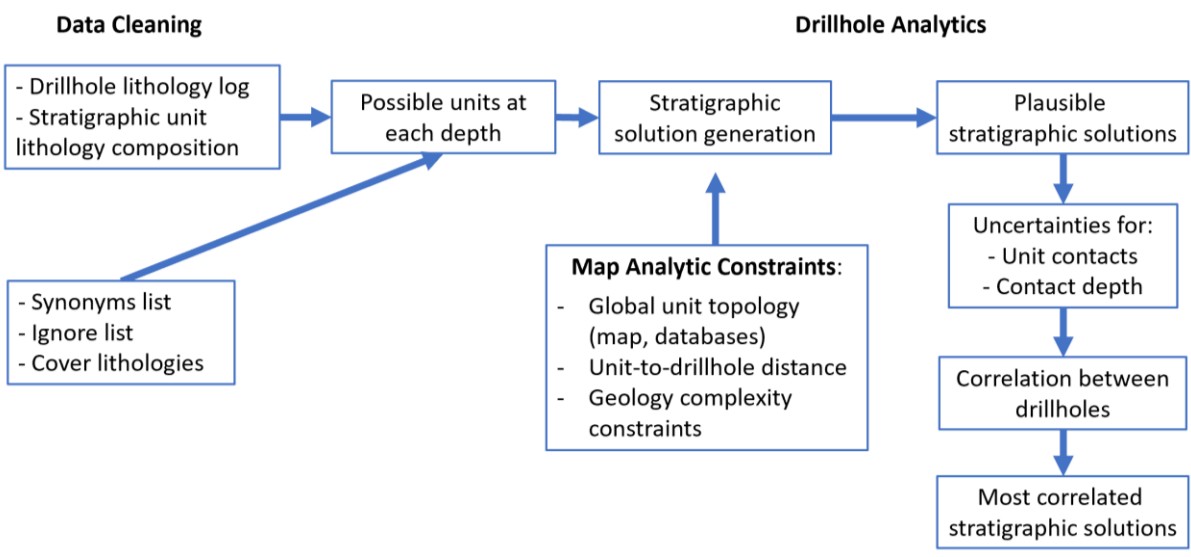

Figure 2: The different stages of the analysis.

The workflow shown in Fig. 2 consists of three key steps grouped into three main tasks: Data Cleaning (using the dh2loop code), Map Analytic Constraints (using map2model and custom codes developed for this project) and Drillhole Analytics (using the litho2strat code developed for this project).

**2.1.1 Data Cleaning**

Prior to analysing the drillhole data we went through a number of automated data cleaning and harmonisation steps.

    a) Harmonisation of drillhole lithology descriptions using the dh2loop code described in (Joshi et al., 2021) (code available here: https://github.com/Loop3D/dh2loop) This enables us to

produce a standardised lithological description for multiple drillholes in a region, regardless of their provenance. This includes the use of a synonym list ("granite" vs "granitoid"), and ignore list (e.g. "fault") together with a list of cover lithology terms (e.g. "saprolite") that enables us to simplify the list of terms and exclude irrelevant information.

b) Harmonisation of lithological descriptions for formations described in the geological map of the target area. This ensures that the same terminology is used for borehole lithological descriptions and map lithologies.

Together steps a and b ensure consistent lithological terminology across drillhole logs and geological map units, enabling subsequent stratigraphic unit matching (Section 2.2)..

### 2.1.2 Map Analytic Constraints

a) Calculation of the distance between each polygon in a map and the target borehole. A custom Python script was developed. This information can be used as a guide to the likelihood that a drillhole would intersect a given unit.

b) We then used the map2model engine (M. Jessell et al., 2021) (code available here: https://github.com/Loop3D/map2model_cpp) to extract the topological relationships between the surface expression of stratigraphic different units. This would later be used to assess the likelihood that two units would be in contact in the drillhole.
The map2model engine extracts topological relationships between stratigraphic units, including both normal depositional contacts and fault contacts, as both types of juxtaposition may be encountered in drillhole data.

Unit connectivity information can also be obtained from the Australian Stratigraphic Units Database (ASUD) as well as from various published reports containing stratigraphic data (Geoscience Australia and Australian Stratigraphy Commission, 2017). The ASUD serves as a comprehensive repository of geological information, providing valuable insights into the relationships between different stratigraphic units across Australia. Additionally, numerous geological surveys and research studies offer stratigraphic data that can further enrich our understanding of unit connectivity. Leveraging this information, enhances stratigraphic models, improves the accuracy of correlations between drillholes, and facilitates a deeper understanding of the geological framework in specific regions.

These two steps allow us to capture information on the spatial and topological relationships between the mapped units.

### 2.1.3 Drillhole Analytics

In this stage, we employ the litho2strat code to generate plausible stratigraphic solutions that fit the observed lithological data while satisfying all geological constraints (code available here: https://github.com/Loop3D/litho2strat; Ogarko et al., 2025). The algorithm uses a recursive branch and prune approach to efficiently explore the solution space, eliminating geologically implausible pathways early in the search process (see Section 2.2 for detailed algorithm description). This strategy not only ensures thorough exploration of viable stratigraphic orderings

but also optimizes computational efficiency by avoiding unnecessary enumeration of invalid solutions.

From the complete ensemble of plausible solutions obtained for each drillhole, we calculate uncertainties that quantify the confidence in different stratigraphic interpretations. Solutions are scored based on the probability of unit contacts within the local solution ensemble, providing a ranking of stratigraphic hypotheses from most to least likely.

To further reduce uncertainty and improve solution reliability, we implement a correlation algorithm that leverages information from multiple neighboring drillholes simultaneously (see Section 2.5 for correlation algorithm details). By comparing the topological relationships of stratigraphic units across drillholes, the correlation process identifies solutions that are geologically consistent across the broader area. Correlated solution scores integrate both local evidence from individual drillholes and regional consistency with neighboring holes, with solutions receiving the highest correlated scores selected as the most plausible stratigraphic interpretations.

## 2.2  Stratigraphic solution generation

The litho2strat algorithm operates through a hierarchical search strategy that systematically explores the space of possible stratigraphic orderings (solutions) while pruning geologically implausible solutions. The algorithm can be formally described as follows:

**Input:**

- $L = \{l_1, l_2, ..., l_n\}$ : sequence of lithologies observed at depths $d_1 < d_2 < ... < d_n$

- $U = \{u_1, u_2, ..., u_m\}$ : set of m candidate stratigraphic units, each defined by its constituent lithologies

- $C$ : set of geological constraints (distance, connectivity, complexity)

- $\Gamma$ : global unit connectivity graph derived from geological maps and stratigraphic databases

**Output:**

- $S = \{s_1, s_2, ..., s_k\}$ : set of k plausible stratigraphic solutions

- $P(s_i)$ : probability distribution over solutions

- $G_h$ : local connectivity graph for drillhole h, derived from all solutions for this drillhole

**Algorithm Steps:**

**1. Unit Matching Phase:** For each lithology $l_i$ at depth $d_i$, identify the subset of compatible units:

$M(l_i) = \{u_j \in U \mid lithology(u_j) \text{ matches } l_i \text{ AND satisfies constraints } C\}$ *(1)*

**2. Recursive Branch and Prune Exploration:** The algorithm recursively builds the solution space from shallow to deep depth intervals. Starting from the surface, partial solutions are extended one depth level at a time by considering candidate units that match the observed lithology. The algorithm generates a new branch for candidate unit $u_j$ only when all of the following conditions are satisfied:

•    The unit $u_j$ matches the observed lithology at the current depth
•    The extended solution satisfies all constraints in $C$ (distance, occurrence, contact complexity)
•    For the last unit $u_k$ in the partial solution, the edge $(u_k, u_j)$ exists in the global connectivity
graph $\Gamma$

Partial solutions that violate any condition are immediately abandoned (pruned), preventing
exploration of their extensions. When a partial solution reaches the deepest depth interval, it is
validated and added to the solution set $S$. This recursive approach with constraint-based pruning
eliminates large portions of the solution space without explicit enumeration.
The algorithm systematically explores all geologically valid solutions through exhaustive search with
constraint-based pruning. While the top-to-bottom traversal order does not affect the completeness
of the final solution set S (the same valid stratigraphic interpretations would be found regardless of
traversal direction), it does improve computational efficiency by enabling earlier application of
surface geology constraints and more effective pruning of invalid solution branches.
**3. Local Connectivity Graph Construction:** From the complete set of solutions $S$ obtained for drillhole
$h$, construct a local connectivity graph $G_h$ where edge weights represent the frequency of unit
contacts across all solutions:
$w_h(u_j, u_{j+1}) = |\{s \in S : (u_j, u_{j+1})\ adjacent\ in\ s\}| / |S|$                 *(2)*
This directed local graph captures the probability of unit contacts based on the ensemble of
geologically plausible solutions for drillhole $h$, where edges represent stratigraphic ordering. Each
edge weight represents the fraction of solutions in which the corresponding unit contact appears.
Note that $G_h$ is a subgraph of the global connectivity graph $\Gamma$, as all solutions for drillhole $h$ must
satisfy the global connectivity constraints.
**4. Solution Scoring:** For each solution $s_i \in S$, calculate a normalized score based on the local
connectivity graph $G_h$:
$score(s_i) = \Sigma_j\ w_h(u_j, u_{j+1}) / N_i$                         *(3)*
where $N_i$ is the number of unit contacts in solution $s_i$ (i.e., $N_i = |s_i| - 1$), and the sum is over all
consecutive unit pairs. The normalization by $N_i$ ensures that solutions with different numbers of
stratigraphic contacts are directly comparable, preventing bias toward longer or more complex
solutions. The score thus represents the average edge probability across all contacts in the solution.
**5. Probability Calculation:** Normalize scores to obtain probability distribution:
$P(s_i) = score(s_i) / \Sigma_k\ score(s_k)$                      *(4)*
The efficiency of this approach derives from constraint-based pruning during the recursive
exploration. By evaluating both solution constraints $C$ and global connectivity $\Gamma$ before extending
each partial solution, the algorithm eliminates inconsistent paths immediately without exploring
their complete extensions. The distinction between the global connectivity graph $\Gamma$ (used for
constraint validation during exploration) and the local connectivity graph $G_h$ (derived from solutions
and used for scoring) is crucial: $\Gamma$ represents *a priori* geological knowledge from maps and databases,
while $G_h$ captures the *a posteriori* probability distribution of unit contacts specific to drillhole $h$ given
all constraints.

## 2.3  Solution constraints

For the Branch and Prune algorithm described in Section 2.2, providing efficient constraints (collectively denoted as $C$) is crucial for generating geologically plausible stratigraphies and reducing the search space. Without these constraints, the algorithm would need to exhaustively enumerate all possible unit assignments, which is computationally prohibitive. We utilize two types of solution constraints: the first can be derived from geological maps (as discussed in the 'Map Analytic Constraints' section), while the second is selected by the user based on the expected structural complexity of the area (e.g., the presence of faults, folds, or other features that might cause stratigraphic repetition or disruption).

**The specific constraints in $C$ include:**

**1. Distance Constraint:** This constraint limits the number of geological units considered based on their proximity to the drillhole. In this context this is defined as the distance between the drillhole collar and the nearest point on the polygon's boundary in 2D. For drillhole $h$ and candidate unit $u_j \in U$:

$$d(u_j, h) \leq dmax, \tag{5}$$

where $d(u_j, h)$ is the distance from the nearest outcrop of unit $u_j$ to drillhole $h$, and $dmax$ is the maximum search radius. This ensures relevance to the drillhole's location.

**2. Global Unit Connectivity Constraint:** This constraint, enforced through the global connectivity graph $\Gamma$, restricts potential contacts between units. For any two consecutive units $u_j$ and $u_{j+1}$ in a solution:

$$(u_j, u_{j+1}) \in E(\Gamma), \tag{6}$$

where $E(\Gamma)$ is the edge set of the global connectivity graph. This ensures that only units known to be stratigraphically adjacent (from map data, databases, or published reports) can be placed in contact, enhancing the geological plausibility of solutions.

The edges in the global connectivity graph $\Gamma$ can be configured as either single-directional or bidirectional depending on the structural complexity of the study area. In structurally simple areas with normal stratigraphic succession, single-directional edges (e.g., A→B) enforce the expected younging direction (older to younger upward). However, for areas with known structural complexities such as overturned sequences from folding or thrust faulting, bidirectional edges can be used to allow stratigraphic contacts in both normal and reversed orientations. For example, if units A and B can occur in both normal succession (A overlies B) and overturned succession (B overlies A) due to folding, the graph $\Gamma$ would include a bidirectional edge between them, allowing transitions in both directions (A→B and B→A). This configuration allows the algorithm to exhaustively explore all structurally valid solutions including those with reversed polarity sequences. The choice of single-directional versus bidirectional edges in $\Gamma$ is thus a key input that controls whether the algorithm considers only normal superposition or also accommodates structural inversions.

**3. Top Unit Constraint:** Information regarding the top unit *utop* can be extracted from geological maps at the surface location of the drillhole, providing a foundational boundary condition:

$$s[0] = utop, \tag{7}$$

where *s[0]* denotes the shallowest unit in solution *s*. Note that while the global unit connectivity constraint allows sequences to begin from any node in the connectivity graph, this constraint explicitly specifies the starting node.

**4. Occurrence Constraint:** This constraint sets a maximum limit on how many times a unit can appear in a solution, accounting for geological complexity such as faulting or folding. For unit $u_j$ in solution $s_i$:

$$count(u_j, s_i) \leq kmax, \tag{8}$$

where *count(u_j, s_i)* is the number of times unit $u_j$ appears in $s_i$. Typically *kmax* = 1 for unfaulted sequences, or *kmax* = 2-3 for faulted terrains where stratigraphic repetition may occur.

**5. Contact Complexity Constraint:** For a continuous sequence of identical lithology observations [$l_i$, $l_{i+1}$, ..., $l_{i+m}$] where all lithologies are the same, this constraint limits the number of distinct stratigraphic units that can be assigned:

$$|\{u_j : assigned\ to\ interval\ [i, i+m]\}| \leq cmax, \tag{9}$$

where *cmax* is the maximum number of unit contacts allowed within the continuous lithology sequence. This prevents over-interpretation where a thick monotonous lithology (e.g., a 100m sandstone sequence) is artificially divided into an excessive number of stratigraphic units.

**6. Stratigraphic Jump Constraint:** To account for incomplete exposure of geological contacts at the surface, we relax the map-based connectivity constraint by allowing the algorithm to "jump" over intermediate units in the global connectivity graph Γ. For a path in Γ such as A→B→C, setting the maximum number of stratigraphic jumps parameter to jmax allows direct contacts between non-adjacent units up to jmax steps apart in the graph. For example, with jmax=1, the algorithm can consider both A→B and A→C as valid contacts, even if A→C is not explicitly observed in the map data. This addresses the limitation that geological maps provide only a 2D surface expression of 3D geological relationships and may not capture all possible stratigraphic contacts that exist at depth. The constraint is defined as:

$$d\Gamma(ui, uj) \leq jmax + 1, \tag{10}$$

where dΓ(ui, uj) is the shortest path distance between units ui and uj in the connectivity graph Γ, and jmax is the maximum number of allowed jumps (typically jmax=0 for strict adherence to observed contacts, or jmax=1-2 for more permissive exploration).

These constraints in *C* work together to enhance the efficiency and effectiveness of the Branch and Prune algorithm, ensuring that the resulting stratigraphies are both geologically plausible and computationally tractable. As demonstrated in Section 3, constraint-based pruning reduces the search space by >99% in practical applications.

## 2.4  Computational complexity

The computational complexity of the branch and prune algorithm depends on several key factors: the number of drillholes H, the length of the lithology sequence |L| (i.e., the number of depth intervals), the number of candidate stratigraphic units |U|, and critically, the average number of solutions N maintained during the recursive exploration. The algorithm processes each drillhole

independently, and for each drillhole, it iterates through all lithologies in L, evaluating potential unit
assignments for each active solution.
The theoretical time complexity can be expressed as:
$O(H \times |L| \times N \times |U|)$,                            (11)
where N denotes the average number of solutions maintained during recursive exploration. This is
the most variable factor and depends strongly on the geological complexity and the constraints
applied.
In the unconstrained case, where no geological constraints are imposed, the number of solutions can
grow exponentially with the number of lithology changes k in the drillhole log, potentially reaching N
$\propto |U|^k$. This leads to a worst-case complexity of $O(H \times |L| \times |U|^{k+1})$, which quickly becomes
computationally prohibitive for complex stratigraphic sequences.
However, the application of geological constraints C - particularly the global unit connectivity
constraint enforced through the topology graph Γ - dramatically reduces the solution space. These
constraints prune geologically implausible branches early in the recursive exploration, preventing
exponential growth of N. In practice, with appropriately chosen constraints, N grows moderately with
the number of lithology changes (approximately linearly rather than exponentially), resulting in
manageable computational requirements even for complex stratigraphic sequences.
The effectiveness of constraint-based pruning in controlling computational cost is demonstrated
empirically in Appendix B, where we compare the growth of average solution numbers as a function
of lithology changes for cases with and without topology constraints.

## 2.5  Solution correlation
We utilize solution correlation analysis to identify compatible stratigraphic orderings between
multiple drillholes, serving as a constraint on the plausibility of individual solutions. This correlation
leverages the topological relationships of units represented through local connectivity graphs from
each drillhole.
A key challenge in correlating stratigraphy logs is that units at the same depth may not align across
different drillholes due to variations in unit dip and thickness, tectonic deformation (including
faulting), and stratigraphic gaps (such as unconformities). To address this, we focus on correlation
based on topological relationships rather than depth-matching. The local connectivity graph $G_h$ for
each drillhole $h$ is constructed from the complete set of solutions $S_h$ obtained via the Branch and
Prune algorithm (Section 2.2), where nodes represent geological units, edges represent stratigraphic
ordering between units, and edge weights $w_h(u_j, u_{j+1})$ (Eq. 2) represent the probability of unit
contacts within that drillhole's solution ensemble.
To facilitate correlation analysis, we generalize the scoring function from Section 2.2 to evaluate any
solution $s_i$ against any local connectivity graph. Define the generalized scoring function as:
$score(s_i, G_h) = \Sigma_j\, w_h(u_j, u_{j+1}) / N_i$,                   (12)
where the sum is over all consecutive unit pairs $(u_j, u_{j+1})$ in solution $s_i$, $G_h$ represents any local
connectivity graph derived from drillhole solutions, $w_h(u_j, u_{j+1})$ denotes the edge weight from graph
$G_h$ for that unit pair, and $N_i$ is the number of unit contacts in solution $s_i$. Note that $G_h$ refers to local
connectivity graphs from drillhole solutions, not the global connectivity graph Γ from map data

(Section 2.2). If an edge $(u_j, u_{j+1})$ from solution $s_i$ does not exist in $G_h$, its weight is taken as zero. This generalized function allows us to assess how consistent a solution from one drillhole is with the geological relationships observed in other drillholes.

**Correlation Algorithm:**

Consider a set of H drillholes {$h_1$, $h_2$, ..., $h\_H$} with their respective local connectivity graphs {$G_1$, $G_2$, ..., $G\_H$}. For each solution $s_i$ from any drillhole, we compute a correlated score that represents the average consistency across all drillholes:

$$scorecorr(s_i) = (1/H) \Sigma_{k=1}^{H} \alpha_k \, score(s_i, G_k), \tag{13}$$

where $\alpha_k$ are weighting factors that can be based on geological distance (distance between collar and closest node of map polygon), drillhole quality, or other criteria. This equation computes an average score across all drillholes. The division by $H$ ensures the correlated score remains on a comparable scale regardless of the number of drillholes. In this work, we use $\alpha_k = 1$ for all drillholes, giving equal weight to each drillhole. This summation approach is robust to outliers; if one drillhole yields a zero score, it does not eliminate the entire correlation. Alternative weighting schemes such as $\alpha_k = 1/d(h_1, h_k)$ could be employed to reduce the influence of more distant drillholes.

The correlated scores are then normalized to obtain a revised probability distribution:

$$Pcorr(s_i) = scorecorr(s_i) / \Sigma_m scorecorr(s_m), \tag{14}$$

The correlated probability $Pcorr(s_i)$ provides a revised ranking of solutions that accounts for both local evidence and regional consistency. Solutions with unit contacts that appear frequently across multiple drillholes receive higher correlated scores, while solutions unique to a single drillhole receive lower scores. This correlation effectively reduces uncertainty by leveraging spatial geological consistency.

**Summation vs. Multiplication:** While the equation for *scorecorr* uses weighted summation, an alternative multiplicative approach could also be formulated. However, multiplicative forms are more sensitive to outliers: if any single drillhole yields a zero score, the entire correlated score becomes zero. Therefore, the summation approach is generally preferred for its robustness.

**Computational Efficiency:** The algorithm achieves $O(H^2 \times S_{avg})$ complexity when correlating solutions across all $H$ drillholes, where $S_{avg}$ represents the average number of solutions per drillhole. This efficiency is achieved by comparing solutions against pre-computed connectivity graphs $G_h$ rather than performing pairwise solution comparisons. The alternative of solution-to-solution comparison would scale as $O(H^2 \times S_{avg}^2)$ making it computationally prohibitive.

By integrating and correlating drillhole data through this topological approach, we ensure that the stratigraphic framework accurately reflects the natural spatial variations and interconnections present in the subsurface. The correlation process quantitatively reduces uncertainty by identifying and favoring solutions that are geologically consistent across the broader area. This uncertainty reduction is achieved by concentrating probability mass on solutions supported by multiple drillholes while downweighting locally anomalous interpretations. The resulting correlated probabilities $Pcorr(s_i)$ provide more reliable stratigraphic interpretations than single-drillhole probabilities $P(s_i)$, enabling more informed decisions in geological exploration and 3D geological modeling.

## 2.6 Code design

A Python package called litho2strat has been developed for stratigraphy recovery. It can be easily installed using the command "pip install", and it has minimal external library dependencies: numpy, matplotlib, and NetworkX. The NetworkX library is utilized to create a directed graph data structure that represents the topological relationships of relative unit ages (Hagberg et al., 2008). It also supports exporting graphs to GML format (Himsolt, 1997) for advanced graph visualization with tools like yEd (https://www.yworks.com/products/yed).

Interaction with the code is facilitated through a *Parfile*, a text file that contains all necessary parameters and paths to the input data files. The parameters in the *Parfile* are organized into several categories based on their functionality, including input file paths, solver settings, and data preprocessing options. An example of such a *Parfile* is provided in Appendix A.

The code architecture efficiently organizes distinct modules, including data reader, the user interface (represented by the *Parfile*), the algorithms (such as the solver), and the visualization components (e.g., output figures and graphs), as shown in Fig. 3. This design enhances code readability, making it easier for developers to understand and navigate the codebase. Additionally, it facilitates further extensions by allowing new features to be integrated seamlessly. This structure also supports effective testing, enabling modifications to be verified systematically and reducing the risk of introducing errors..

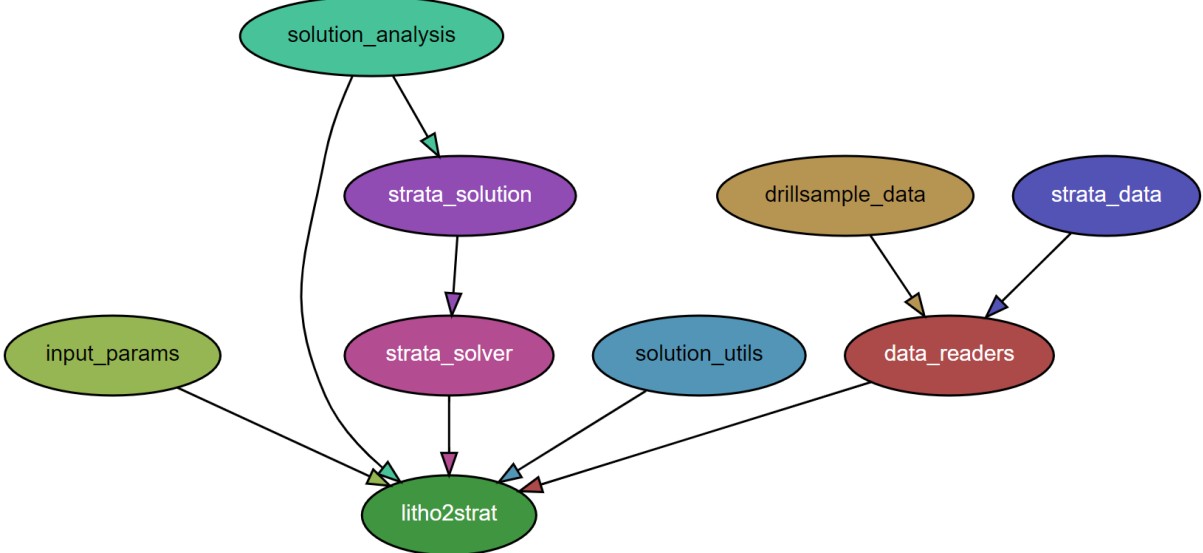

Figure 3: The module dependencies of the *litho2strat* code. The graph is generated by the *pydeps* utility, while excluding external dependencies.

## 3. Example Use

For this example, we used a set of 52 drillholes from South Australia originally drilled by Teck Cominco Pty. Ltd. (Fig. 4). This area was chosen as there were a number of holes equally spaced with a relatively homogenous spatial distribution and the holes provided both lithological logs and existing interpretations of the down-hole stratigraphy.

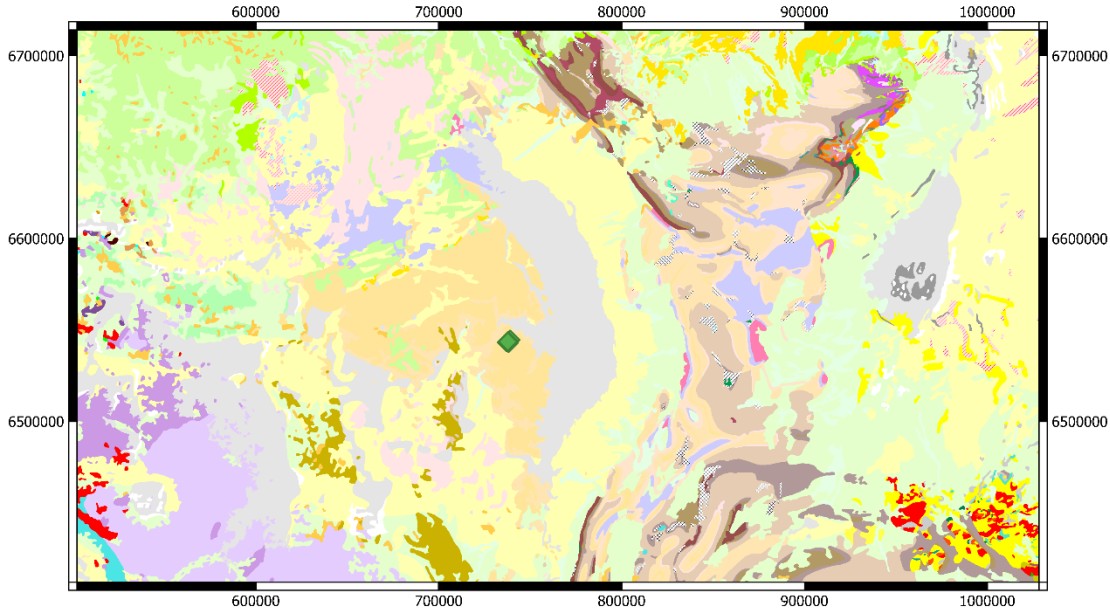



Figure 4: Location of South Australia test area (drillholes shown as green diamonds), together with an
example stratigraphic log, map from 1:2M Surface Geology Map of South Australia (The Department
for Energy and Mining, the Government of South Australia, Geoscientific. Data, Sourced on 22 July
2018, http://energymining.sa.gov.au/minerals/geoscience/geological_survey/data GDA94/Zone 53).
**Data Cleaning**

Examples of terms in the ignore list for this case study include the following, where each term is
excluded from drillhole lithology log processing:
1. Breccia (Undiff. Origin)
2. Ironstone (Metasomatic)
3. No Information
4. Solution-Collapse Breccia
5. Vein (Undifferentiated)

Examples of the thesaurus of synonyms for this case study area include the following groups, where
each group contains lithology names that are treated as equivalent:
1. dolomite, dolomite rock, carbonate rock, limestone
2. conglomerate, diamictite
3. grit, sandstone, quartzite, siltstone
4. gabbro, gabbronorite

**Map Analytics**
Figure 5 shows stratigraphic units coloured as a function of the distance to one of the drillholes. A
large search area was used for this example as the stratigraphy is fairly flat lying so there is no
guarantee that a unit will reach the surface in the local neighbourhood.

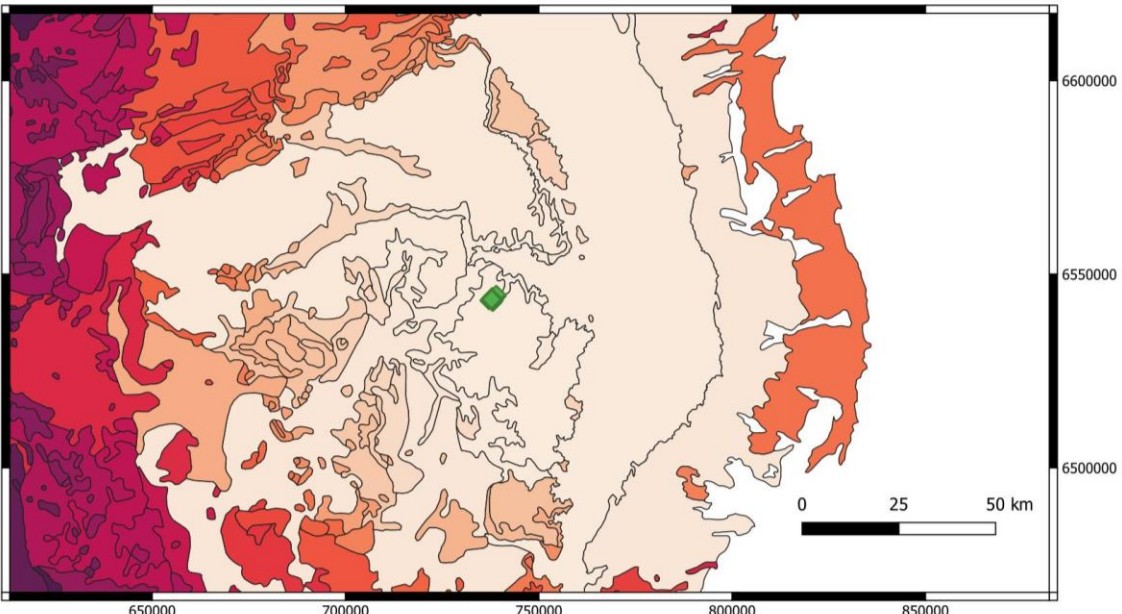


Figure 5. Distance of stratigraphic units from drillholes (darker colours signifies larger distance).
Green diamonds show the location of the drillholes (Same source map as Fig. 4, GDA94/Zone 53).
In the initial analysis we constructed the global connectivity graph Γ (Section 2.2), representing
topological relationships between stratigraphic units. The initial graph was constructed automatically
from the geological map (extending out 100 km from the test area) using the map2model software,
then manually extended with additional topological relationships from the ASUD database and
published reports. The graph was processed using the NetworkX Python library, exported to GML
format, and visualized using yEd software (Fig. 6). The global connectivity graph consists primarily of
single-direction edges, with two bidirectional edges (Whyalla Sandstone–Angepena Formation and
Paleoproterozoic-Mesoproterozoic Rocks–Donington Suite) to account for spatial variability in their
stratigraphic relationships.

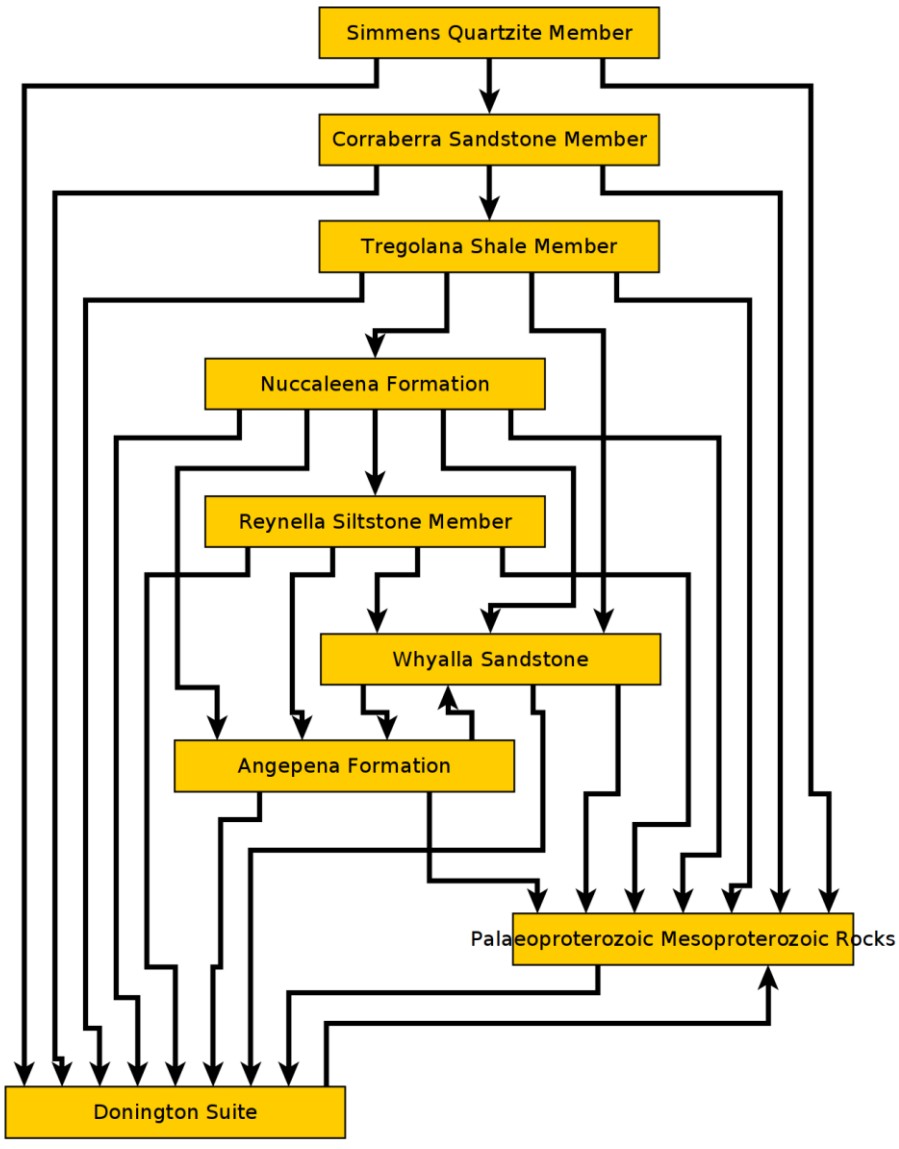


Figure 6: Topological relationships between units in and around the test area.




**Drillhole Analytics**


The drillhole analysis calculated every possible stratigraphic ordering that was consistent with the


observed lithological ordering down the drillhole and solution constraints (described in Sec. 2.3). By


collating the results for all possible solution paths, we can produce estimates of the marginal


probability that any depth interval will be a particular stratigraphic unit (Fig. 7). For depth interval i


and stratigraphic unit u, the probability P_i(u) is computed as:


$$P_i(u) = |\{s \in S : s[i] = u\}| / |S|, \qquad\qquad (15)$$


where S is the set of all valid solutions and s[i] denotes the unit assigned to interval i in solution s.




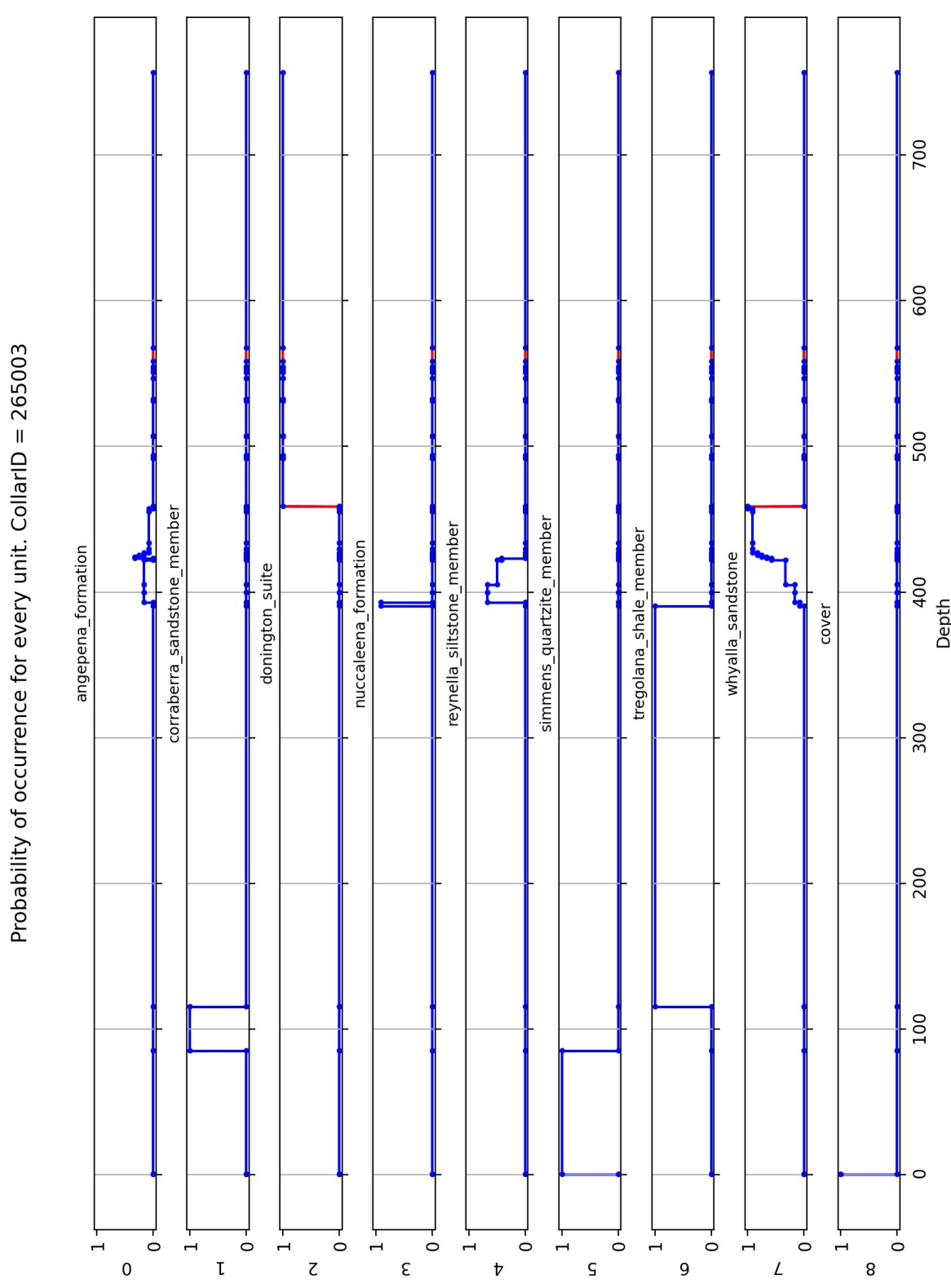

Figure 7: Estimated probability of each stratigraphic unit occurring at a given depth for a single drillhole.

In Fig. 8, we present the final (local) unit connectivity derived from the stratigraphic solutions generated. The width of the graph edges indicates the probability of unit contacts, with thicker edges

signifying higher probabilities. This visual representation allows for a clear comparison of
connectivity before (Fig. 6) and after the stratigraphic analysis.

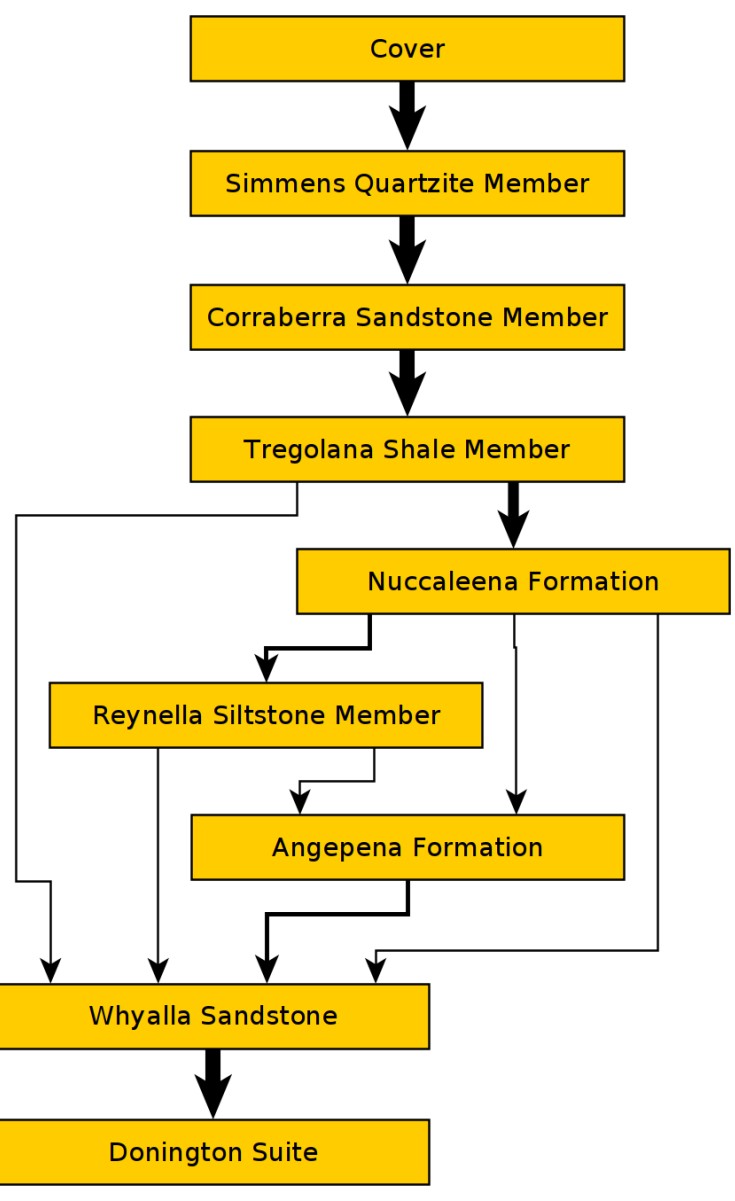


Figure 8: Calculated local topology using all solutions. Graph edges (relationships) between two
stratigraphic units are displayed as a probability of a that contact-relationship occurring.

The final solution score for a single ordering is calculated by summing of the probabilities of the
contact edge weights. This allows us to sort the orderings by probability, ignoring stratigraphic
thickness for now (Fig. 9).

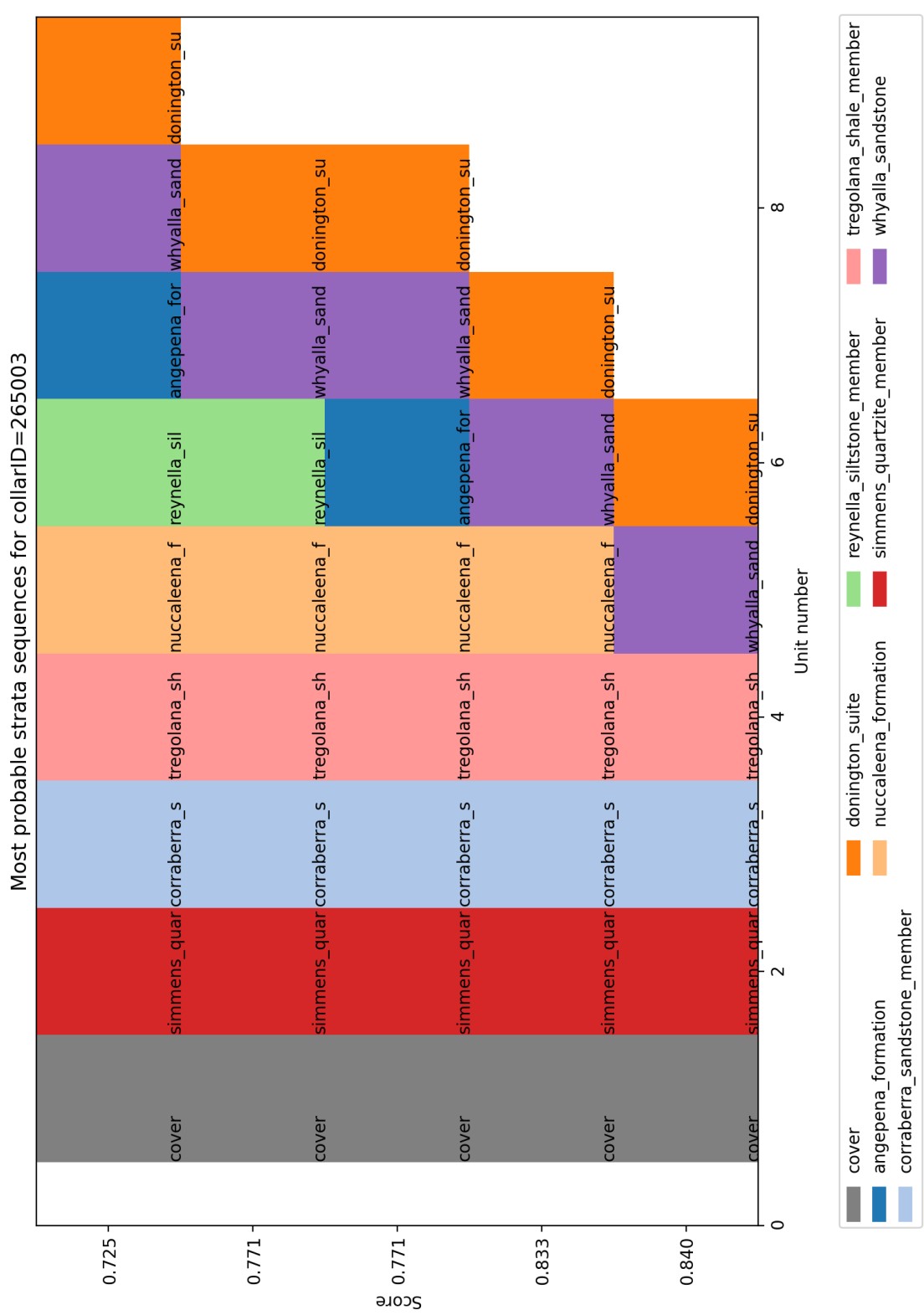


Figure 9: The 5 most probable stratigraphic orderings, with their solution probability on the x axis and order of depth on the y axis.



Finally, we can then include the depths to contacts between units in the drillhole based on the
previous analyses (Fig. 10).

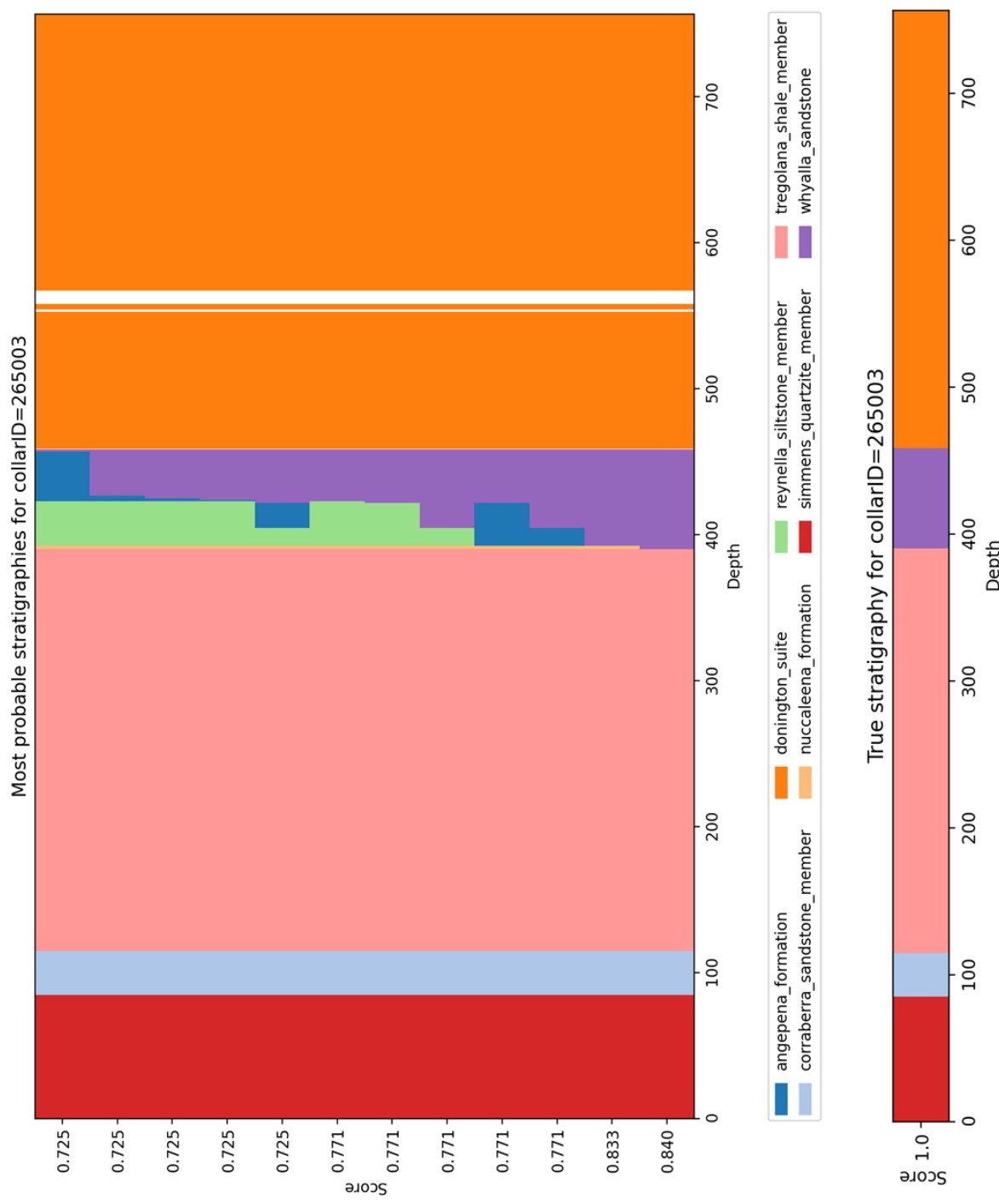


Figure 10: The 12 most probable stratigraphic orderings showing true depth of contact (above)
compared to the stratigraphy as logged for the same hole.


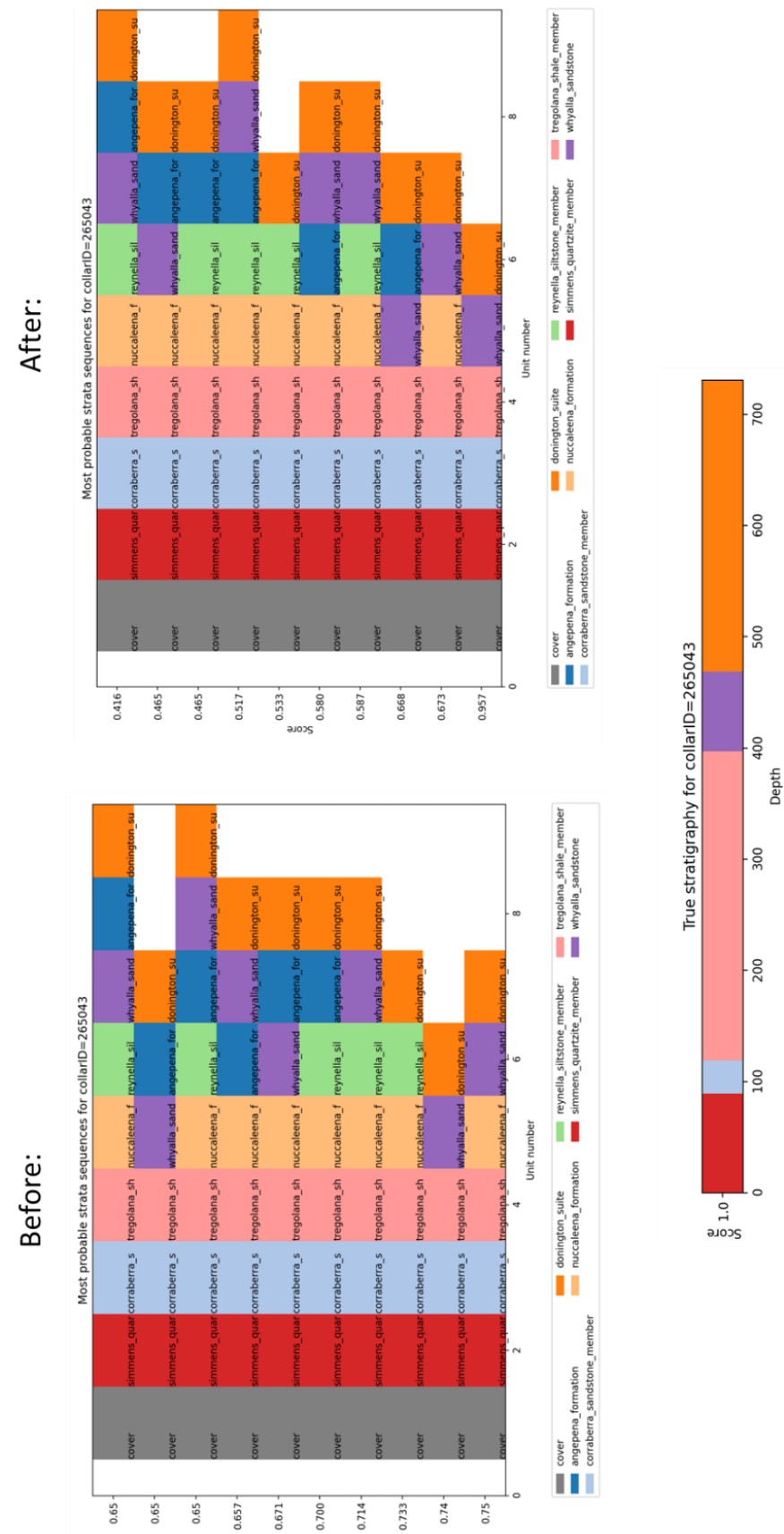


Figure 11: Comparison of ordering for one hole (left) vs ordering for that hole considering the
outcomes of 45 other drillholes in the neighbourhood.

In the next stage of our analysis, we perform solution correlation across multiple drill holes to establish a plausible stratigraphic order and reduce uncertainty. Figure 11 illustrates the comparison of the most probable stratigraphies before and after correlation. Prior to correlation, the solution that aligns with the "true" stratigraphy (the correct solution) is ranked second, with a score of S=0.74, while the highest-ranked solution has a score of S=0.75. However, after applying the correlation, the correct solution rises to the top rank with a score of S=0.95, whereas the previously highest-ranked solution falls to second place with a score of S=0.67. This correlation analysis not only helped identify the correct solution but also significantly reduced its relative uncertainty, increasing the relative score between the top two solutions from 1% to 42%.

The computational efficiency of the litho2strat algorithm was evaluated through performance testing on this dataset, with scalability analysis presented in Appendix B.

# 4. Discussion and Future Work

Whilst we were able to develop a workflow that successfully provided useful stratigraphic analyses for our test area, we recognise that for other areas the methodology was not always as successful. We have identified several aspects of the current stratigraphic descriptions that we think will significantly expand the useability of the workflow we present above.

1) Lithological Uncertainty. Vague lithological descriptions are a major limitation. In many areas, the lithological descriptions of stratigraphic units are quite vague, and successive stratigraphic units in a group might have very similar lithological descriptions..

   As an example, we look at the Hamersley Group, in Western Australia (Maldonado & Mercer, 2018). If we examine the GSWA explanatory notes for three successive formations (Mt McRae Shale, Mt Sylvia Formation and the Wittenoom Formation) in the GSWA explanatory notes their lithologies are described as:

   - **Mt McRae Shale** - Mudstone, siltstone, chert, iron-formation, and dolomite. Thin bands of shard-bearing volcanic ash in upper parts.
   - **Mt Sylvia Formation** - Mudstone, siltstone, chert, iron-formation, and dolomite.
   - **Wittenoom Formation** - Thinly bedded dolomite and dolomitic shale, with minor black chert, shale, banded iron formation and sandstone.

   We can see that there is a significant overlap in lithologies, with an ordering of lithologies but without constraints on the percentage of each lithology in the three formations. This additional information, even as an estimate, would provide useful constraints on the likelihood that a specific lithology is associated with a given stratigraphic unit.

2) Min-Max thickness estimates. In some areas, there is useful information on the minimum, maximum and average stratigraphic thickness of units.

3) Stratigraphic ordering of lithologies. Additional information on commonly occurring orderings of lithologies within a given formation or member would also provide useful constraints.

## West Angela Member

### Derivation of name/Formal lithostratigraphy

The West Angela Member was the first subdivision of the Wittenoom Formation to be formally recognized (Blockley et al., 1993). It is named after West Angela Hill (Zone 50, MGA 673387E 7442407N) near the West Angelas iron ore mine, and the type section is defined as the interval between 420.4 m and 524.6 m in drill hole WRL 1 (Blockley et al., 1993) stored at the Geological Survey of Western Australia (GSWA) Carlisle Core Library.

Five shaly horizons separated by BIF, chert, or massive dolomite are recognized in the West Angela Member and are informally designated as AS1 to AS5 (Kepert, 2018). In particular the lower three shale horizons form a distinctive pattern in natural gamma-ray logs that can be used for regional correlation (Blockley et al., 1993).

*Minimum thickness* (m)      —
*Maximum thickness* (m)      80

### Lithology

The West Angela Member is generally not well-exposed and consists predominantly of dolomite and shaly dolomite, with minor chert, BIF, volcaniclastic rocks, and impact ejecta layers. Near the base, there is a distinctive unit of interbedded chert, BIF, dolomitic shale, and shale with characteristic natural gamma-ray peaks that are designated AS1 to AS3 (Blockley et al., 1993). This entire interval is referred to as A1 by some mining companies (e.g. Kepert, 2018) and is overlain by a thick interval of shale and dolomitic shale (AS3). The middle of the member, between AS3 and AS 4, contains a unit of massive to laminated crystalline dolomite with local carbonaceous shale and siltstone partings (Blockley et al., 1993). The upper part of the West Angela Member (AS4 to AS5) consists mainly of dolomitic shale and shale with minor chert beds that is gradationally overlain by massive dolomite at the base of the Paraburdoo Member. Lateral correlations between drillholes WRL 1 and FVG 1 suggest that the member becomes shalier towards the east (Blockley et al., 1993).

Figure 12: Free-text descriptions of the West Angela Member in the GSWA Explanatory Notes.

All three of these types of information are often included in the free-text portions of stratigraphic databases, such as the example shown for the West Angela Member in the GSWA Explanatory Notes in Fig. 12. In this example the free text provides more specific information on the thickness, the ordering of lithologies and the relative proportions of lithologies. With the advent of more sophisticated Machine Learning methodologies, the extraction of this ancillary data in a standardised form from reports and the stratigraphic databases themselves will open up new possibilities for constraining stratigraphy. Similarly, the codes developed in dh2loop for harmonising lithological terminologies will expand greatly in coming years.

4) Inclusion of discontinuity information in the litho2strat workflow (most often logged faults) could help to define where breaks in stratigraphy are most likely to occur

5) Inclusion of secondary descriptive information (for example grain size) could help to refine our younging estimators in areas of uncertain facing.

6) There is no doubt that the advent of Large Language Models will have a profound effect on our ability to extract and categorize information from unstructured data sources, and algorithms based on these approaches will probably replace the data extraction and data harmonisation modules in future versions of this workflow.

# 5. Conclusions



We developed codes and methodologies for stratigraphy recovery from drillhole databases, utilizing
the branch and prune algorithm as a foundational framework. To ensure the generation of
geologically plausible solutions, we implemented various types of constraints that account for the
complexities of subsurface geology.
To further reduce uncertainty in the obtained solutions, we introduced a correlation algorithm that
leverages information from multiple drillholes simultaneously. This innovative approach allows for a
more robust analysis by integrating data across different locations, enhancing the reliability of the
stratigraphic interpretations.
Our proposed method was applied to a dataset comprising 52 drillholes from South Australia. The
results demonstrated that the algorithm successfully predicts the correct stratigraphic solution while
providing associated uncertainty metrics, effectively validating its performance against measured
stratigraphy data.
Additionally, we identified several key aspects of the current stratigraphic descriptions that could
significantly enhance the usability of the workflow we have presented. These enhancements aim to
improve the accessibility and applicability of our methodology, paving the way for more effective
geological assessments and decision-making processes in the field.

















*Code and data availability.* The software and datasets used in this study are publicly available for
download at GitHub (https://github.com/Loop3D/litho2strat) and Zenodo
(https://doi.org/10.5281/zenodo.15064469, Ogarko et al., 2025).
*Author contribution.* VO and MJ are the primary contributors to this study. VO led the research,
developed the methodology and software, and prepared the manuscript. MJ provided guidance on
drillhole data analysis and contributed to manuscript writing.
*Competing interests.* The authors declare that they have no conflict of interest.
*Acknowledgements*. The work has been supported by the Mineral Exploration Cooperative Research
Centre whose activities are funded by the Australian Government's Cooperative Research Centre
Program. This is MinEx CRC Document 2025/27. We are grateful to reviewer Guillaume Caumon for
very helpful comments that greatly improved the paper.

# Appendix A- Control file for litho2strat code



Example usage: python3 litho2strat.py -p ./parfiles/Parfile_SA.txt
Example parfile:

```
[FilePaths]
topology_filename = data/SA_test_data/newpairs_20_06_2023.gml
ignore_list_filename = data/SA_test_data/ignore_list.txt
alternative_rock_names_filename = data/SA_test_data/alternative_rock_names.txt
unit_colors_filename = data/SA_test_data/unit_colors.csv
drillsample_filename = data/SA_test_data/litho_tables/litho_$collarID$.csv
stratasample_filename = data/SA_test_data/strat_tables/strat_$collarID$.csv
dist_table_filename = data/SA_test_data/dh_asud_strat2.csv
[DataHeaders]
drillsample_header = DEPTH_FROM_M, DEPTH_TO_M, MAJOR_LITHOLOGY,
stratasample_header = DEPTH_FROM_M, DEPTH_TO_M, STRAT_UNIT_NAME,
strata_data_header = strat, summary, distance, description
[SolverParameters]
add_topology_constraints = True
max_num_strata_jumps = 0
max_num_returns_per_unit = 0
max_num_unit_contacts_inside_litho = 0
single_top_unit = True
[DataPreprocessing]
number_nearest_units = 10
min_drillhole_litho_score = 80
group_drillhole_lithos = False
cover_ratio_threshold = 0.65
[CollarIDs]
collarIDs = 205821,205822,264999,265000,265001
```

# Appendix B: Performance and Scalability Analysis

To complement the theoretical complexity analysis presented in Section 2.4, we conducted empirical tests to evaluate the performance and scalability of the litho2strat algorithm. We tested how the average number of solutions maintained during recursive exploration (N) scales with the number of lithology changes in drillhole logs, comparing two scenarios: (1) using the global topology graph Γ as a constraint, and (2) without topology constraints.

Figure B.1 shows the relationship between the number of lithology changes and the average number of solutions maintained during recursive exploration when the topology graph constraint is applied. The results demonstrate near-linear scaling, confirming that the topology graph effectively prunes the solution space while preserving geological validity.

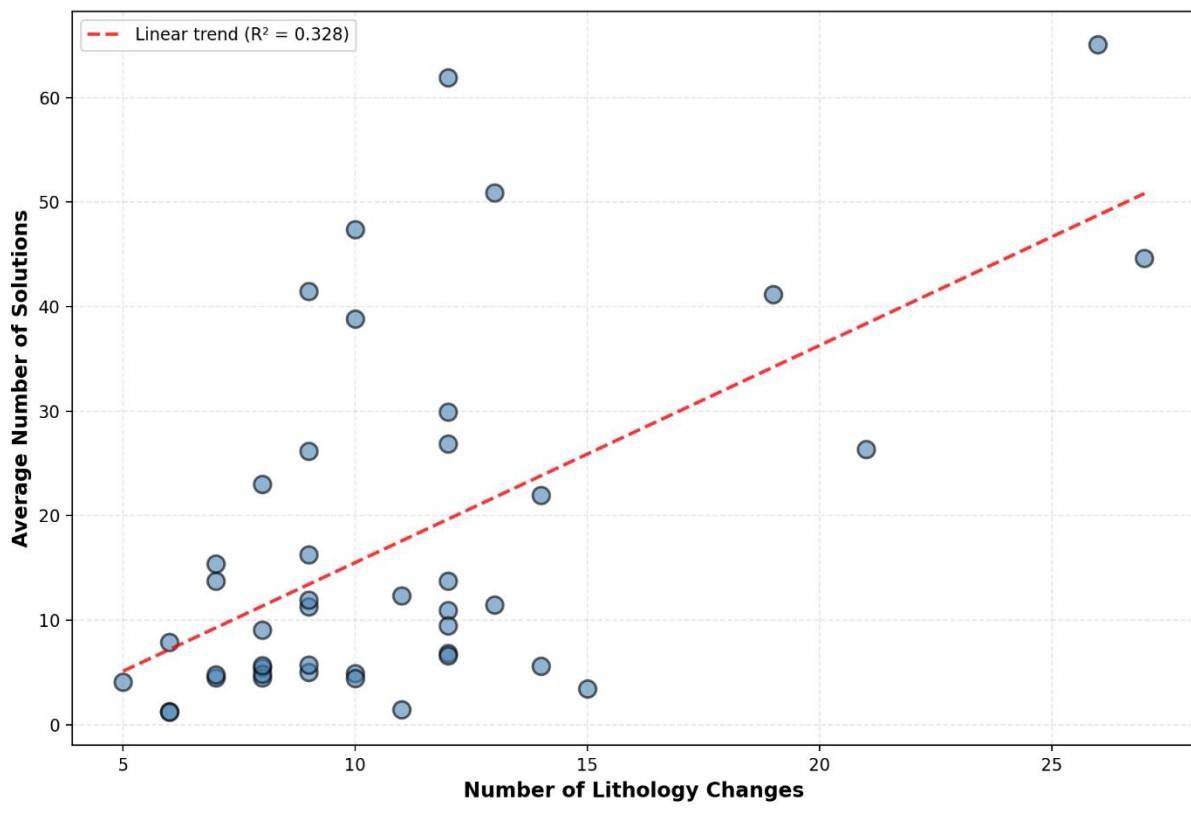

Figure B.1: Average number of solutions maintained during recursive exploration versus number of lithology changes with topology graph constraint.

Figure B.2 presents the same relationship for the unconstrained case, where the algorithm considers all theoretically possible stratigraphic interpretations. Here, the average number of solutions maintained during recursive exploration exhibits near-exponential growth with increasing lithology changes, illustrating the combinatorial explosion that occurs without geological constraints.

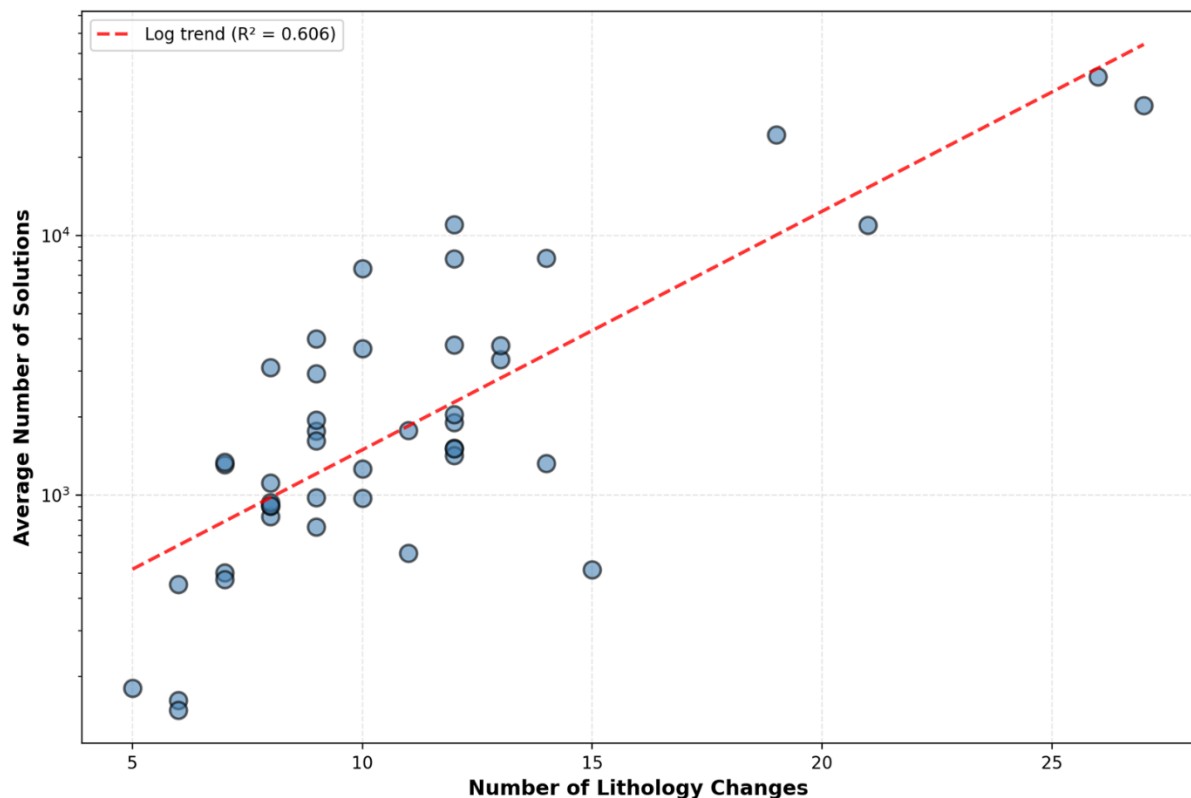

Figure B.2: Average number of solutions maintained during recursive exploration versus number of lithology changes without topology constraints.

The computational performance measurements further highlight the practical importance of these constraints. Using a single CPU core (Intel i7-1185G7 @ 3.00GHz) to process all 52 drillholes from Section 3 and perform the correlation of solutions, the constrained approach required approximately 1 second total processing time, while the unconstrained case required approximately 50 seconds for the same dataset. This 50-fold improvement in computational efficiency, combined with the near-linear versus near-exponential scaling behavior of solutions maintained during recursive exploration, demonstrates that incorporating geological knowledge through the topology graph is essential for both computational tractability and practical applicability of the litho2strat algorithm to real-world geological datasets.


Alvarado-Neves, F., Ailleres, L., Grose, L., Cruden, A. R., & Armit, R. (2024). Three-dimensional
geological modelling of igneous intrusions in LoopStructural v1.5.10. *Geoscientific Model*
*Development*, *17*(5), 1975–1993. https://doi.org/10.5194/gmd-17-1975-2024
Calcagno, P., Chilès, J. P., Courrioux, G., & Guillen, A. (2008). Geological modelling from field data and
geological knowledge. *Physics of the Earth and Planetary Interiors*, *171*(1–4), 147–157.
https://doi.org/10.1016/j.pepi.2008.06.013
Caumon, G., Collon-Drouaillet, P., Le Carlier de Veslud, C., Viseur, S., & Sausse, J. (2009). Surface-
Based 3D Modeling of Geological Structures. *Mathematical Geosciences*, *41*(8), 927–945.
https://doi.org/10.1007/s11004-009-9244-2
D'Affonseca, F. M., Finkel, M., & Cirpka, O. A. (2020). Combining implicit geological modeling, field
surveys, and hydrogeological modeling to describe groundwater flow in a karst aquifer.
*Hydrogeology Journal*, *28*(8), 2779–2802. https://doi.org/10.1007/s10040-020-02220-z
Fullagar, P.K., Zhou, B., and Biggs, M., 2004**.** Stratigraphically consistent autointerpretation of
borehole data. Journal of Applied Geophysics, 55(1-2), 91-104.
https://doi.org/10.1016/j.jappgeo.2003.06.010
Geoscience Australia and Australian Stratigraphy Commission (2017). Australian Stratigraphic Units
Database. https://www.ga.gov.au/data-pubs/datastandards/stratigraphic-units
Giraud, J., Pakyuz-Charrier, E., Jessell, M., Lindsay, M., Martin, R., & Ogarko, V. (2017). Uncertainty
reduction through geologically conditioned petrophysical constraints in joint inversion.
*GEOPHYSICS*, *82*(6), ID19–ID34. https://doi.org/10.1190/geo2016-0615.1
Guo, J., Wang, Z., Li, C., Li, F., Jessell, M. W., Wu, L., & Wang, J. (2022). Multiple-Point Geostatistics-
Based Three-Dimensional Automatic Geological Modeling and Uncertainty Analysis for
Borehole Data. *Natural Resources Research*, *31*(5), 2347–2367.
https://doi.org/10.1007/s11053-022-10071-6
Guo, J., Xu, X., Wang, L., Wang, X., Wu, L., Jessell, M., Ogarko, V., Liu, Z., & Zheng, Y. (2024). GeoPDNN
1.0: a semi-supervised deep learning neural network using pseudo-labels for three-dimensional
shallow strata modelling and uncertainty analysis in urban areas from borehole data.
*Geoscientific Model Development*, *17*(3), 957–973. https://doi.org/10.5194/gmd-17-957-2024
Hagberg, A. A., Schult, D. A., & Swart, P. J. (2008). *Exploring Network Structure, Dynamics, and*
*Function using NetworkX*. 11–15. https://doi.org/10.25080/TCWV9851
Hartmann, J., & Moosdorf, N. (2012). The new global lithological map database GLiM: A
representation of rock properties at the Earth surface. *Geochemistry, Geophysics, Geosystems*,
*13*(12). https://doi.org/10.1029/2012GC004370
Hill, E.J., Pearce, M.A. & Stromberg, J.M. Improving Automated Geological Logging of Drill Holes by
Incorporating Multiscale Spatial Methods. *Math Geosci* 53, 21–53 (2021).
https://doi.org/10.1007/s11004-020-09859-0
Himsolt, M. (1997). *GML: a portable graph file format*.
Jessell, M. (2001). Three-dimensional geological modelling of potential-field data. *Computers &*
*Geosciences*, *27*(4), 455–465. https://doi.org/10.1016/S0098-3004(00)00142-4

Jessell, M., Aillères, L., Kemp, E. de, Lindsay, M., Wellmann, F., Hillier, M., Laurent, G., Carmichael, T., & Martin, R. (2014). Next Generation Three-Dimensional Geologic Modeling and Inversion. In *Building Exploration Capability for the 21st Century*. Society of Economic Geologists. https://doi.org/10.5382/SP.18.13

Jessell, M., Ogarko, V., de Rose, Y., Lindsay, M., Joshi, R., Piechocka, A., Grose, L., de la Varga, M., Ailleres, L., & Pirot, G. (2021). Automated geological map deconstruction for 3D model construction using &lt;i&gt;map2loop&lt;/i&gt; 1.0 and &lt;i&gt;map2model&lt;/i&gt; 1.0. *Geoscientific Model Development*, *14*(8), 5063–5092. https://doi.org/10.5194/gmd-14-5063-2021

Jessell, M. W., Ailleres, L., & de Kemp, E. A. (2010). Towards an integrated inversion of geoscientific data: What price of geology? *Tectonophysics*, *490*(3–4), 294–306. https://doi.org/10.1016/j.tecto.2010.05.020

Joshi, R., Madaiah, K., Jessell, M., Lindsay, M., & Pirot, G. (2021). &lt;i&gt;dh2loop&lt;/i&gt; 1.0: an open-source Python library for automated processing and classification of geological logs. *Geoscientific Model Development*, *14*(11), 6711–6740. https://doi.org/10.5194/gmd-14-6711-2021

Land, A. H., & Doig, A. G. (1960). An Automatic Method of Solving Discrete Programming Problems. *Econometrica*, *28*(3), 497. https://doi.org/10.2307/1910129

Lindsay, M. D., Jessell, M. W., Ailleres, L., Perrouty, S., de Kemp, E., & Betts, P. G. (2013). Geodiversity: Exploration of 3D geological model space. *Tectonophysics*, *594*, 27–37. https://doi.org/10.1016/j.tecto.2013.03.013

Maldonado, A., & Mercer, K. (2018). *Comparison of the Laboratory and Barton-Bandis Derived Shear Strength of Bedding Partings in Fresh Shales of the Pilbara, Western Australia*. *All Days*, ISRM-ARMS10-2018-204.

Mallet, J.-L. L. (2002). *Geomodeling*. Oxford University Press, Inc.

Martin, R., Ogarko, V., Giraud, J., Plazolles, B., Angrand, P., Rousse, S., & Macouin, M. (2024). Gravity data inversion of the Pyrenees range using Taguchi sensitivity analysis and ADMM bound constraints based on seismic data. *Geophysical Journal International*, *240*(1), 829–858. https://doi.org/10.1093/gji/ggae410

Ogarko, V., Giraud, J., Martin, R., & Jessell, M. (2021). Disjoint interval bound constraints using the alternating direction method of multipliers for geologically constrained inversion: Application to gravity data. *GEOPHYSICS*, *86*(2), G1–G11. https://doi.org/10.1190/geo2019-0633.1

Ogarko, V., & Jessell, M. (2025). litho2strat 1.0 source code, https://doi.org/10.5281/zenodo.15064469

Pakyuz-Charrier, E., Giraud, J., Ogarko, V., Lindsay, M., & Jessell, M. (2018). Drillhole uncertainty propagation for three-dimensional geological modeling using Monte Carlo. *Tectonophysics*, *747–748*, 16–39. https://doi.org/10.1016/j.tecto.2018.09.005

Schetselaar, E. M., & Lemieux, D. (2012). A drill hole query algorithm for extracting lithostratigraphic contacts in support of 3D geologic modelling in crystalline basement. *Computers & Geosciences*, *44*, 146–155. https://doi.org/10.1016/j.cageo.2011.10.015

Silversides, K., Melkumyan, A., Wyman, D.A., and Hatherly, P., 2015. Automated recognition of
stratigraphic marker shales from geophysical logs in iron ore deposits. Computers &
Geosciences, 77, 118-125. https://doi.org/10.1016/j.cageo.2015.02.002

Tarantola, A. (2005). *Inverse Problem Theory and Methods for Model Parameter Estimation*. Society
for Industrial and Applied Mathematics. https://doi.org/10.1137/1.9780898717921

Vollgger, S. A., Cruden, A. R., Ailleres, L., & Cowan, E. J. (2015). Regional dome evolution and its
control on ore-grade distribution: Insights from 3D implicit modelling of the Navachab gold
deposit, Namibia. *Ore Geology Reviews*, *69*, 268–284.
https://doi.org/10.1016/j.oregeorev.2015.02.020

Wedge, D., Hartley, O., McMickan, A., Green, T., and Holden, E., 2019. Machine learning assisted
geological interpretation of drillhole data: Examples from the Pilbara Region, Western Australia.
Ore Geology Reviews, 114, 103118. https://doi.org/10.1016/j.oregeorev.2019.103118

Wellmann, F., & Caumon, G. (2018). *3-D Structural geological models: Concepts, methods, and
uncertainties* (pp. 1–121). https://doi.org/10.1016/bs.agph.2018.09.001

Wu, X., and Nyland, E., (1987). Automated stratigraphic interpretation of well-log data. Geophysics,
52(12), 1665-1676. https://doi.org/10.1190/1.1442283
