# Peer review of "Automated stratigraphic interpretation from drillhole lithological descriptions with uncertainty quantification: litho2strat 1.0"

_EGUsphere, 2025_

## Author Comment (AC1)

**Recovery of stratigraphic data with associated uncertainties from drillhole databases using litho2strat 1.0**

**Author Responses to Referee Comments**

Vitaliy Ogarko and Mark Jessel

We thank Guillaume Caumon for the comprehensive and constructive review of our manuscript. His feedback and constructive comments have greatly improved the clarity of the manuscript. Below, we provide responses to all referee comments and include information on updates made to the manuscript.

**Main remarks:**

1. In terms of form, the paper would benefit from a better problem statement and more precise wording. I was a bit in the fog when I started reading because I was not very clear about the input, output and overall objective of the method. Things became clearer when moving forward, as details and examples were given. A reason for this is terminology, which is sometimes very general (please see some suggestions on this in the detailed remarks below). The flowchart of Fig. 1 is useful, and could be moved to the introduction. Adding some some visual examples in the flowchart would help, and explaining what is done in geological terms would be needed at this stage (more than how it is done --"optimization solver"-- or using a different type of frame). Adding some visuals could also support definitions of terms in the introduction.

We thank the referee for these constructive suggestions to improve the clarity and accessibility of our paper. We have made several significant additions to address these concerns:

1. **Visual Example in Introduction:** We have added Figure 1a to the Introduction, which provides a clear visual example of the fundamental problem. This figure illustrates how a single drillhole log containing only lithological descriptions (sandstone, siltstone, sandstone, dolomite) can yield multiple plausible stratigraphic interpretations. The accompanying text explains in geological terms how the same lithology may represent different stratigraphic formations, either repeated by faulting or belonging to distinct units with similar compositions. This concrete example immediately clarifies the challenge our method addresses.

2. **Clear Problem Statement:** We have added a comprehensive paragraph in the Introduction that formally defines:

   o **Inputs:** (1) legacy drillhole databases with lithological descriptions, (2) regional geological maps defining stratigraphic unit boundaries, and (3) topological constraints from relative ages and depositional sequences

   o **Outputs:** (1) stratigraphic unit assignments for each depth interval, (2) multiple plausible solutions ranked by geological likelihood, and (3) quantified uncertainties

   o **Objectives:** We clearly articulate the threefold objective of transforming lithological descriptions into stratigraphic interpretations, quantifying uncertainties, and establishing correlations between drillholes

3. **Geological Context:** The new problem statement explicitly explains why this transformation is essential: "regional 3D geological models are fundamentally organized by stratigraphy rather than

lithology, yet the majority of legacy drillhole data lack stratigraphic labels." This provides the geological motivation before diving into technical details.

4. **Improved Terminology:** Throughout the revised sections, we have been more precise with terminology, using specific geological terms and clearly defining technical concepts when first introduced. For example, we now explicitly refer to "stratigraphic solutions" and "stratigraphic unit assignments" to clarify what our algorithm produces, rather than using generic terms.

5. **Enhanced Workflow Visualization:** We have revised the workflow diagram (Figure 1) with clearer terminology as suggested. The diagram now specifies "Stratigraphic solution generation," shows Map Analytic Constraints including global unit topology from maps and databases, clarifies "Unit-to-drillhole distance," and distinguishes between intermediate and final outputs (plausible stratigraphic solutions and correlated solutions).

We believe these additions, particularly the visual example and formal problem statement early in the Introduction, provide readers with a clear understanding of the geological problem and our approach before encountering the technical methodology. The flowchart (Figure 1) remains in the Methodology section where it can be understood in the context of the detailed workflow description.

Would it be correct to state that the proposed method does a clustering of lithofacies data with topological (adjacency) constraints?

While there are superficial similarities, our method is not clustering. Key distinctions: (1) We assign predefined stratigraphic units to depth intervals rather than discovering clusters in the data; (2) We systematically explore all valid stratigraphic orderings using branch-and-prune optimization, not grouping similar lithofacies; (3) Our topological constraints represent known geological relationships (relative ages, depositional sequences) rather than data-driven adjacencies. The method is better characterized as "constrained stratigraphic interpretation through systematic hypothesis testing" - closer to a constraint satisfaction problem than clustering.

2. Also on the form, the paper could describe a bit more completely some of the methods used. This includes the overall algorithm / type of distance (Line 96 --is this a map distance or 3D distance?); the basic algorithms and input of the litho2strat code (Line 118), in particular the branch and bound algorithm. A supporting figure and example would help. I appreciate that the code is provided, but a higher level description in the paper with a few more equations would certainly help. A more formal description of the uncertainty quantification would also help. (please see detailed remarks).

We thank the referee for highlighting the need for more complete and formal descriptions of our methods. We have substantially expanded the methodology sections to address these concerns:

1. **Supporting Figure and Example:** We have added Figure 1a to the Introduction, which provides a schematic illustration of the stratigraphic interpretation problem. This figure shows how a drillhole log containing only lithological descriptions can yield multiple plausible stratigraphic solutions, clearly demonstrating why our methodology is needed and setting up the conceptual framework for the litho2strat algorithm described in the paper.

2. **Algorithm Description (Section 2.2):** We have added a new section "Stratigraphic solution generation" that provides a comprehensive mathematical description of the litho2strat algorithm. This includes:

- o Formal specification of inputs (lithology sequences, candidate stratigraphic units, constraints, and connectivity graphs)

- o Detailed algorithm steps with mathematical notation

- o Description of the recursive branch-and-prune exploration strategy

- o Mathematical formulation of solution scoring and probability calculations

- o Clear explanation of how the algorithm constructs local connectivity graphs from solution ensembles

3. **Solution Constraints (Section 2.3):** We have completely rewritten this section to provide mathematical definitions for each constraint type, including:

- o Distance constraints (with clarification that this refers to distance between the drillhole collar and the nearest point on the polygon's boundary in 2D)

- o Global unit connectivity constraints

- o Occurrence and contact complexity constraints

- o Each constraint is now formally defined with mathematical notation and clear explanations

4. **Solution Correlation (Section 2.5):** We have rewritten this section with formal mathematical descriptions of:

- o The generalized scoring function for evaluating solutions against connectivity graphs

- o The correlation algorithm with step-by-step mathematical formulations

- o Uncertainty quantification through probability distributions derived from correlated scores

- o Computational complexity considerations

These additions provide the higher-level description with equations and visual support that the referee requested, making the methodology more accessible without requiring readers to examine the code. We believe these formal descriptions and the illustrative figure significantly improve the reproducibility and clarity of our approach.

3. Research gaps: the motivation for the method with regard to previous literature should be made more explicitly after line 60: what are the problems with the (many) well data classification approaches that motivate this contribution?

Thank you for highlighting the need to better position our work relative to existing approaches. After reviewing the literature on automated drillhole interpretation, we identified that most existing methods address a fundamentally different problem than ours:

**Existing Approaches and Their Limitations:**

1. **Geophysics-to-Stratigraphy Methods**: The majority of automated interpretation methods use geophysical logs as primary inputs to predict stratigraphic units (Wu & Nyland, 1987 - using gamma, resistivity, and porosity logs; Fullagar et al., 2004 - wireline log interpretation; Silversides et al., 2015 - using natural gamma logs to identify marker shales). These approaches require either distinctive

geophysical signatures that may not exist in all geological settings, or extensive training datasets of pre-interpreted drillholes.

2. **Geochemical/Spectral Methods**: Other approaches use specialized data such as geochemical assays, XRF scanning, or hyperspectral measurements to identify geological boundaries (Hill & Uvarova, 2018). While powerful, these require expensive data collection that is not available in legacy drillhole databases.

3. **Hybrid Machine Learning Approaches**: Some recent methods combine multiple data types, such as Wedge et al. (2019) who use lithological composition, assays, and geophysical logs from the Pilbara iron ore deposits. However, these approaches primarily use lithology as training data for machine learning classifiers rather than directly interpreting lithological descriptions, and require extensive pre-interpreted drillhole datasets for training.

**The Gap We Address:**

Critically, these methods do not solve the problem faced by geological surveys worldwide: millions of legacy drillholes contain only lithological descriptions (e.g., "sandstone," "siltstone," "dolomite") but lack both stratigraphic interpretations and geophysical logs. Our method specifically addresses this gap by:

- **Using lithological descriptions as the sole required input**, without needing geophysical or geochemical measurements

- **Requiring no training data** - instead leveraging existing geological knowledge from maps and stratigraphic databases

- **Systematically exploring all geologically plausible interpretations** through branch-and-prune optimization

- **Quantifying uncertainty** inherent in the lithology-to-stratigraphy transformation

- **Correlating between multiple drillholes** to reduce uncertainty

To our knowledge, this is the first method designed specifically to transform lithological descriptions from legacy drillhole databases into multiple ranked stratigraphic interpretations with quantified uncertainties, without requiring geophysical logs or pre-interpreted training data. This addresses a critical need for utilizing the vast amounts of lithology-only drillhole data held by geological surveys globally. We have added a paragraph to the introduction to better clarify the scope and novelty of our approach.

4. Representativity of map contacts: as maps provide a section view of the 3D medium, there is not guarantee that geological maps provide an exhaustive set of possible contacts between stratigraphic units. While I appreciate the interest of constraining the solution space only to the observed contacts, I fear that this could also lead to under-estimating the actual uncertainties in many cases (and I guess this can be easily addressed). Please discuss.

We thank the referee for this important observation about the limitations of using geological map contacts as constraints. We fully agree that geological maps provide only a 2D surface expression of 3D geological relationships and may not capture all possible stratigraphic contacts that exist at depth, potentially leading to under-estimation of uncertainties.

We note that our methodology incorporates connectivity information not only from geological maps but also from stratigraphic databases and published reports (as described in Section 2.1.2), which often contain subsurface information derived from drillholes, seismic surveys, and other data sources. This multi-source approach already provides some constraints on stratigraphic relationships beyond what is visible at the surface.

Additionally, to further address this concern, we have implemented a "Stratigraphic Jump Constraint" (now described as Constraint 6 in Section 2.3) that specifically handles cases where the available data may not capture all possible contacts. This constraint allows the algorithm to consider contacts between stratigraphic units that are not directly connected in the observed topology but are geologically plausible based on their stratigraphic proximity.

The constraint works by permitting the algorithm to "jump" over intermediate units in the global connectivity graph $\Gamma$. For example, if the available data shows a sequence A→B→C, setting the jump parameter $jmax=1$ allows the algorithm to also consider direct A→C contacts that might occur but are not documented. This approach maintains geological plausibility while acknowledging that even combined surface and subsurface data may not be exhaustive.

In practice, users can adjust the $jmax$ parameter based on the data completeness and geological complexity of their study area: $jmax=0$ for areas with comprehensive data coverage and simple geology, or $jmax=1-2$ for areas with limited data or known structural complexity. This flexibility ensures that uncertainty estimates appropriately reflect both the available data and its limitations.

5. **Along these lines, does the code make a distinction between "normal" stratigraphic contacts and stratigraphic contacts due to fault juxtaposition?**

Thank you for this important question. We have clarified in Section 2.1.2 that the map2model software does identify both normal stratigraphic contacts and fault contacts from geological maps. We retain both types of contacts in the connectivity graph $\Gamma$, as fault juxtapositions represent valid unit relationships that are commonly encountered in drillholes. The algorithm therefore can handle both normal stratigraphic successions and fault-juxtaposed units, with the occurrence constraint (Section 2.2) allowing for stratigraphic repetitions where faulting is present.

6. **It seems that the used algorithm is greedy, so that it does not depend on the well traversal order (top to bottom or conversely); is this correct?**

Thank you for this clarification question. The referee is correct - our branch and prune algorithm exhaustively explores all valid solutions (it is not greedy), so the traversal order does not affect the completeness of the final solution set S. The same geologically valid stratigraphic interpretations would be found whether traversing top-to-bottom or bottom-to-top.

We chose top-to-bottom traversal for computational efficiency: it enables earlier pruning through surface geology constraints and aligns with natural geological interpretation workflow. While the algorithm could work in either direction, the top-to-bottom approach provides better pruning opportunities in practice.

We have added clarification in Section 2.2 explicitly stating that the algorithm performs exhaustive search and explaining why top-to-bottom traversal was chosen for efficiency while noting it doesn't affect solution completeness.

7. Does the proposed method use polarity? There is a mention to relative stratigraphic ages and directed graphs in the code design section, but polarity is not discussed before. In the end, I am not sure if the method handles only ordered sequences (like Dynamic time warping) or not. The aggregation of results to define the stratigraphic succession seems to preclude reverse series due to reverse faults or folds. Please clarify and discuss.

Thank you for this important question about stratigraphic polarity. Our method CAN handle reversed stratigraphic sequences (due to folding or reverse faulting) through appropriate configuration of the global topology graph Γ.

**How polarity is handled:**

- The global topology graph Γ (derived from geological maps, ASUD database, and stratigraphic reports - Section 2.2) defines which unit contacts are geologically valid.

- This graph can be configured as bidirectional to allow contacts in both normal and reversed orientations.

- For areas with known structural complexity (overturned sequences, recumbent folds), bidirectional edges between units allow the algorithm to explore both normal (A overlies B) and reversed (B overlies A) sequences.

- The algorithm will then exhaustively find all valid solutions including those with reversed polarity.

- In our South Australian case study, the global connectivity graph consists primarily of single-direction edges, with two bidirectional edges (Whyalla Sandstone–Angepena Formation and Paleoproterozoic Mesoproterozoic Rocks–Donington Suite) to account for spatial variability in their stratigraphic relationships.

  **Clarification added to manuscript:** We have added text to Section 2.2 clarifying that the topology graph can be bidirectional to handle reversed sequences, and that polarity handling is controlled by the graph configuration based on the expected structural complexity of the study area. We have also added a note in Section 3 indicating that the global connectivity graph contains two bidirectional edges.

8. Could you provide a few elements about performance and scalability of the code?

We thank the referee for this important suggestion. We have comprehensively addressed the performance and scalability aspects of our code through the following additions:

**1. Theoretical Complexity Analysis (New Section 2.4):** We added a new section on "Computational Complexity" that provides a formal analysis of the branch and prune algorithm's theoretical time complexity. We demonstrate that the complexity is $O(H \times |L| \times N \times |U|)$, where H is the number of drillholes, $|L|$ is the length of the lithology sequence, N is the average number of solutions maintained during recursive exploration, and $|U|$ is the number of candidate stratigraphic units. We explain how N is the critical factor: geological constraints dramatically reduce it from exponential growth in the unconstrained case to near-linear growth when topology constraints are applied.

**2. Correlation Algorithm Efficiency (Section 2.5):** We expanded Section 2.5 to include analysis of the correlation algorithm's computational efficiency. We show that the algorithm achieves $O(H^2|S|)$ complexity when correlating solutions across all H drillholes, where $|S|$ is the size of the solution set.

This efficiency is achieved by using pre-computed connectivity graphs rather than direct solution-to-solution comparison, which would scale as $O(H^2|S|^2)$. This design choice makes the correlation computationally tractable even for large datasets.

**3. Empirical Performance Testing (New Appendix B):** We added a comprehensive Appendix B presenting empirical performance measurements on our South Australian dataset (52 drillholes). The appendix includes:

- Figure B.1 showing near-linear scaling of solution numbers with lithology changes when using topology constraints

- Figure B.2 demonstrating near-exponential growth without constraints

- Quantitative performance metrics: ~1 second total processing time with constraints versus ~50 seconds without constraints (50-fold improvement) on a single CPU core (Intel i7-1185G7 @ 3.00GHz)

These additions demonstrate that the litho2strat algorithm is both theoretically sound and practically efficient, with constraint-based pruning reducing the search space by >99% in real-world applications while maintaining geological validity.

**Detailed remarks:**

- 14: "There are many more wells drilled in the search for oil and shallower holes related to hydrogeology": sentence seems odd. Please rephrase.

    We have revised the sentence to read: "*In addition to mineral exploration drilling, extensive drillhole datasets exist from oil and gas exploration and hydrogeological studies*."

- Line 16 of the abstract also mentions drillhole data, but the algorithms takes lithofacies as input, and does not currently allow for logging, geochemical or assay data, so being more specific would help.

    We have revised line 16 to be more specific: "*Together these legacy drillhole datasets have the potential to significantly improve our subsurface data coverage but have limited use as constraints on regional 3D geological models as many if not most drill logs lack stratigraphic information, containing only lithological descriptions*." We have also clarified in the next paragraph that our method specifically recovers stratigraphy from "drillhole lithological data" rather than the more general "drillhole databases".

- Line 20 mentions "a correlation algorithm", but this appears to be only a step of the overall workflow, and not doing exactly the same thing as stratigraphic correlation as for instance in Waterman and Raymond (1987). summarizing the other steps and explaining what is correlated with what would help to disambiguate the term.

    Thank you for this important clarification. We have revised the abstract to clarify that our "solution correlation algorithm" compares the topological relationships of stratigraphic units across multiple drillholes to identify geologically consistent solutions. We have also added the description of the solution search algorithm step to better explain the overall workflow.

- 22: "integrating uncertainty quantification and presenting multiple geological hypotheses": this is written as two distinct ideas, but I think the scenarios is the way to perform the uncertainty quantification here. Also, a more specific term about the nature of the hypotheses could be relevant.

We have revised the text to read: "*In addition, the method quantifies uncertainty by generating multiple plausible stratigraphic interpretations, providing critical insights for resource estimation, scenario analysis, and data acquisition strategies.*"

- 38: "complexly coded lithological information but limited stratigraphic data": Please disambiguate. What does "complexly coded" mean: lithologies coded as integers or something else? We understand later in the paper that textual descriptions are used. This would be worth explaining from the onset. Also, I'd argue that lithological information is part of lithostratigraphic data, so please replace "stratigraphic data" by "stratigraphic formations / units" (or a more relevant term).

We have revised the text to read: "*GSOs' databases typically contain lithological information as unstructured text descriptions (e.g., 'sandy limestone with minor shale') but rarely include stratigraphic unit assignments.*"

- 42: check references data base (M Jessell et al. should read Jessell et al).

We adjusted the reference.

- 44 "From these" : among these?

You are right, we corrected this.

- 50-52: The last sentence is correct, but it diverts the reader from the focus of the paper. Could be removed or moved to the discussion or conclusion.

We removed this sentence.

- 55: "stratigraphy": I suggest to replace by "stratigraphic units" here and in other places of the manuscript

We have reviewed uses of "stratigraphy" versus "stratigraphic units" and made adjustments where appropriate. We retain "stratigraphy" where it refers to the ordered sequence of units and use "stratigraphic units" when referring to the specific formations themselves.

- 61 "stratigraphy recovery": please define / explain. "drillhole databases"; "data from multiple drillholes": please be more specific. "we enhanced the robustness and reliability of stratigraphic interpretations": enhanced with regard to what ? Please check syntax of the whole sentence. Probably present is better than preterit in this paragraph.

We have revised the text to read: "*This study develops open-source codes and methodologies for stratigraphy recovery (determining the ordered sequence of stratigraphic units) from drillhole lithological data by introducing a search algorithm that systematically explores all geologically plausible stratigraphic orderings for individual drillholes, combined with a solution correlation algorithm that compares the topological relationships of stratigraphic units across multiple drillholes to identify geologically consistent solutions and reduce uncertainty.*" We have also updated the corresponding paragraph in the Introduction with similar clarifications.

- 66-70: reads more like an abstract or a conclusion than an introduction.

We have revised the text to read: "*The method quantifies uncertainty by generating multiple plausible stratigraphic interpretations, providing critical insights for resource estimation, scenario analysis, and data acquisition strategies. We apply our method to a dataset of 52 drillholes from South Australia to demonstrate its practical application and validate its performance against existing stratigraphic interpretations.*"

- **1: what do "ASAD" and "geology complexity" mean?**

  Thank you for requesting this clarification. By "geological complexity" we specifically mean structural features that can disrupt normal stratigraphic succession, such as faults (causing repetition or omission of units), folds (potentially creating overturned sequences), or unconformities. We have clarified this in the text by specifying "structural complexity" and providing explicit examples: "the presence of faults, folds, or other features that might cause stratigraphic repetition or disruption."

  We also corrected ASAD to ASUD and defined it as: "Australian Stratigraphic Units Database".

- **110: "our": 1st person is not useful here.**

  We have revised this sentence to remove the first-person pronoun.

- **120 "Combinatorial optimisation solver": for what problem exactly?**

  We have revised this to specify that we use a branch-and-prune approach to efficiently explore the solution space. Additionally, we have added Section 2.2, which provides a detailed explanation of the algorithm.

- **151-154: Is the top unit constrain a specific case of the global unit connectivity?**

  These are distinct constraints. The global unit connectivity constraint defines the permissible contacts between units but does not specify where in the graph the stratigraphic sequence begins - it can start from any node. The top unit constraint explicitly specifies the starting node, thereby restricting which unit appears at the top of the sequence. This distinction has been clarified in Section 2.2.

- **168: "relationships": what type of relationships?**

  We have revised the text to read: "*We utilize solution correlation analysis to identify compatible stratigraphic orderings between multiple drillholes, serving as a constraint on the plausibility of individual solutions.*"

- **173: tectonic features + stratigraphic gaps may also lead to misalignments.**

  We have revised the text to read: "*A key challenge in correlating stratigraphy logs is that units at the same depth may not align across different drillholes due to variations in unit dip and thickness, tectonic deformation, and stratigraphic gaps (such as unconformities or erosional surfaces).*"

- **174: what do the nodes / edges of the connectivity graph represent? Is the graph oriented or not?**

  We have revised the text to read: "*The local connectivity graph $G_h$ for each drillhole h is constructed from the complete set of solutions $S_h$ obtained via the Branch and Prune algorithm (Section 2.2), where nodes represent geological units, edges represent stratigraphic ordering between units, and edge weights $w_h(u_j, u_{j+1})$ represent the probability of unit contacts within that drillhole's solution ensemble.*" We also added details on the local connectivity graph construction to Section 2.2.

- **181: Give the equation for the solution score.**

  We added the equation.

- **185: what is an "external drillhole"?**

  We have eliminated the "external drillhole" terminology and reformulated the solution correlation approach to be more general and mathematically rigorous. Rather than distinguishing between a target drillhole and "external" ones, the revised algorithm treats all H drillholes uniformly, computing

a correlated score for each solution as the average consistency across all drillholes. The complete revised description appears in Section 2.5.

- 206 "geological distance": please define.

In this context the distance is calculated as the distance between the drillhole collar and the nearest point on the polygon's boundary in 2D. We clarified this in the paper.

- 229: "ensuring that modifications can be verified without introducing errors.": please rephrase.

We have revised the text to read: "*This structure also supports effective testing, enabling modifications to be verified systematically and reducing the risk of introducing errors.*"

- Fig. 3: please add coordinates. Why is there a score on he drillhole ?

We have updated the Figure and added coordinates.

- 249: "The figure below shows": please use figure number

We added the figure number.

249: distance between drillhole and polygon: is this the minimum, maximum or average distance?

In this context the distance is calculated as the distance between the drillhole collar and the nearest point on the polygon's boundary in 2D.

- Check caption of Fig. 4

We corrected the figure caption.

- 5: Would it make sense to replace topological by adjacency? It this graph created automatically?

We retained 'topological relationships' rather than 'adjacency' because the graph is directed and captures stratigraphic ordering (i.e., which unit overlies which), not just spatial adjacency. The initial graph was constructed automatically from the geology map (extending out 100 km from the test area) using the map2model software, then manually extended with additional topological relationships from the ASUD database and published reports. We have clarified this in the revised text.

- 267: is the probability marginal or conditional to the adjacent units? Some equations would help.

The probabilities are marginal - they are computed independently of the adjacent units. For each depth interval i and stratigraphic unit u, the probability $P_i(u)$ represents the fraction of valid solutions in which unit u appears at interval i: $P_i(u) = |\{s \in S : s[i] = u\}| / |S|$, where S is the set of all valid solutions. We have added this equation and clarification to the revised manuscript.

---

## Author Response (AR2)

**Authors' Response to Reviewer Comments**

We thank the reviewer for their thorough and constructive feedback on our revised manuscript. We greatly appreciate the positive assessment of our revisions and are pleased that the changes have improved the clarity of the paper. We have carefully addressed all the minor remarks raised in this review, as detailed below. We believe these final corrections have further strengthened the manuscript.

Detailed remarks:
- 18: "Improve our subsurface data coverage": with regard to what ? Consider rephrasing.

We rephrased the sentence to: "*Together these legacy drillhole datasets have the potential to significantly enhance constraints on regional 3D geological models and improve our understanding of subsurface architecture, but have limited use in their current form as many if not most drill logs lack stratigraphic information, containing only lithological descriptions.*"

- 46: "Drillhole data serves" : serve ?

Yes, we corrected it.

- 165: could be appropriate to also reference a stable url (DOI).

We added a reference to ASUD with an url.

- 174: This sentence is unclear at this stage, and the process become clearer in Sec. 2.2 Maybe add a reference to that section?

We changed this sentence to: "*Together steps a and b ensure consistent lithological terminology across drillhole logs and geological map units, enabling subsequent stratigraphic unit matching (Section 2.2).*"

- 285: Adding an equation number could make it easier to reference this equation in Section

We added equation numbers.

- 387-394: The first sentence of this paragraph is a bit ambiguous (the reader needs to guess that the problem is on the map contacts). Mention that this concerns the connectivity constraint.

We change the sentence to: "*To account for incomplete exposure of geological contacts at the surface, we relax the map-based connectivity constraint by allowing the algorithm to "jump" over intermediate units in the global connectivity graph $\Gamma$.*"

- 402: the term "in seconds" is vague and may be incorrect for huge data bases. Given the following nice section on computational complexity, I suggest removing it at this stage of the paper.

We removed that part, as suggested.

- 417: Use exponent notation.

We applied the exponent notation.

- 483: "unconformities or erosion surfaces": erosions surfaces are unconformities.

We removed "erosion surfaces".

- 488: referencing the w_h equation would help.

We added a reference to the w_h equation.

- 522: Reminding the |S| is the number of solutions would be good. "n drill holes" should read H drillholes.

We adjusted the sentence to: "*The algorithm achieves $O(H^2 \times S_{avg})$ complexity when correlating solutions across all H drillholes, where $S_{avg}$ represents the average number of solutions per drillhole.*"

604: "geology maps": geological maps?

We corrected this.

670 "for this" unclear to me.

We changed that part to: "*Vague lithological descriptions are a major limitation. In many areas, the lithological descriptions of stratigraphic units are quite vague, and successive stratigraphic units in a group might have very similar lithological descriptions.*"

We have also revised the manuscript title to better reflect the methodology and contributions of our work. The new title is: 'Automated stratigraphic interpretation from drillhole lithological descriptions with uncertainty quantification: litho2strat 1.0'. We believe this more accurately conveys the automated nature of the approach and the focus on uncertainty quantification.